# Channel cross-section heterogeneity of particulate organic carbon transport in the Huanghe

Yutian Ke[1][†], Damien Calmels[1], Julien Bouchez[2], Marc Massault[1], Benjamin Chetelat[3], Aurélie Noret[1], Hongming Cai[2], Jiubin Chen[3], Jérôme Gaillardet[2], Cécile Quantin[1]

[1]GEOPS, Université Paris-Saclay-CNRS, 91405 Orsay, France
[2]Université de Paris, Institut de Physique du Globe de Paris, CNRS, 75005 Paris, France
[3]School of Earth System Science, Institute of Surface-Earth System Science, Tianjin University, 300072 Tianjin, China
[†]Present address: Division of Geological and Planetary Science, California Institute of Technology, Pasadena, CA 91125, USA

*Corresponding author*: Yutian Ke (yutianke@caltech.edu)

**Abstract.** The Huanghe (Yellow River), one of the largest turbid river systems in the world, has long been recognized as a major contributor of suspended particulate matter (SPM) to the ocean. However, over the last few decades, the SPM export flux of the Huanghe has decreased over 90% due to the high management, impacting the global export of particulate organic carbon (POC). To better constrain sources and modes of transport of POC beyond the previously investigated transportation of POC near the channel surface, SPM samples were for the first time collected over a whole channel cross-section in the lower Huanghe. Riverine SPM samples were analyzed for particle size and major element contents, as well as for POC content and dual carbon isotopes ($^{13}$C and $^{14}$C). Clear vertical and lateral heterogeneities of the physical and chemical properties of SPM are observed within the river cross-section. For instance, finer SPM carry in general more POC with higher $^{14}$C activity near the surface of the right bank. Notably, we discuss how bank erosion in the alluvial plain is likely to generate lateral heterogeneity in POC composition. The Huanghe POC is millennial-aged (4,020 ± 500 radiocarbon years), dominated by organic carbon (OC) from the biosphere, while the lithospheric fraction is ca. 12%. The mobilization of aged and refractory OC, including radiocarbon-dead biospheric OC, from deeper soil horizons of the loess-paleosol sequence through erosion in the Chinese Loess Plateau is an important mechanism contributing to fluvial POC in the Huanghe drainage basin. Altogether, anthropogenic activities can drastically change the compositions and transport dynamics of fluvial POC, consequentially altering the feedback of the source-to-sink trajectory of a river system to regional and global carbon cycles.

## 1 Introduction

Rivers are the main conveyor of rock and soil debris eroded from the continents to the ocean. Along with inorganic material, river sediments host particulate organic carbon (POC) derived mainly from three major sources: 1) recently photosynthesized OC of the biosphere, 2) aged and altered OC from soils, and 3) ancient OC contained in sedimentary rocks (Blair et al., 2010). The net effect of riverine POC transport on the carbon cycle and thus on the evolution of Earth's climate depends on POC provenance and fate. The effective sedimentary burial of POC derived from the terrestrial biosphere (biospheric OC, $OC_{bio}$) represents a net, long-term sink of atmospheric $CO_2$ (Galy et al., 2007, Bouchez et al., 2014; Hilton et al., 2015), whereas the oxidation of POC derived from continental rocks (petrogenic OC, $OC_{petro}$) acts as a net, long-term source of $CO_2$ to the atmosphere (Hilton et al., 2014). The erosion and burial of $OC_{petro}$ escaping from oxidation has no net effect on the long-term carbon cycle (Galy et al., 2008a; Bouchez et al., 2010; Hilton et al., 2011; Horan et al., 2019). In addition, the reactive nature of $OC_{bio}$ might also result in short-term $CO_2$ emission during transport from both river channels and recently-deposited sediments (Mayorga et al., 2005; Galy and Eglinton, 2011; Blair and Aller, 2012).

Globally, rivers transport a total POC flux of *ca*. 200 Tg C/year, consisting of $157^{+74}_{-50}$ Tg C/year of $OC_{bio}$ and $43^{+61}_{-25}$ Tg C/year of $OC_{petro}$ (Galy et al., 2015; Ludwig et al., 1996). Source-to-sink processes controlling the origin and fate of riverine POC are prominently river-specific, suggesting that the impact of POC on regional and global carbon cycles might significantly vary both spatially and temporally (Blair and Aller, 2012). It is thus crucial to understand the mechanisms controlling the POC export by large rivers that integrate vast portions of the land surface, and quantify the differing sources of carbon exported by those large river systems.

The Huanghe (Yellow River) is a highly turbid river system that exports over 85% of its OC as particulate matter, with efficient deposition and preservation in the ocean (Cauwet and Mackenzie, 1993; Bianchi, 2011; Zhang et al., 2013; Ran et al., 2013). The Huanghe has been highly managed over the last few decades through water and soil conservation measures as well as reservoir construction, leading to a decrease of nearly 90% of its sediment load (Wang and Fu et al., 2016; Wang et al., 2007; Milliman et al., 1987) and a significant decrease in its POC delivery to the ocean (Zhang et al., 2013). Reservoir construction dramatically affects the transport and fate of both sediment load and POC in large rivers (Syvitski et al., 2005; Li et al., 2015). The estimated POC flux of the Huanghe is thought to have shifted from 4.5 Tg C/yr in the 1980s (Cauwet and Mackenzie, 1993) to 0.34-0.58 Tg C/yr nowadays (Tao et al., 2018) in response to both anthropogenic influence (Hu et al., 2015; Tao et al., 2018; Yu et al., 2019a) and natural variability of the regional hydrological cycle (Qu et al., 2020). These large-scale perturbations have likely modified the OC input from the different terrestrial pools as well as the fate of exported POC that was previously reaching deposition centers in the ocean and that now remains stuck on land. Those alterations of the carbon cycle remain to be addressed.

Over the last decade, POC transport in the Huanghe has been investigated for 1) determination and quantification of POC sources, based on bulk or molecular carbon isotopic composition (Tao et al., 2015; Yu et al., 2019b; Ge et al., 2020; Qu et al., 2020); 2) temporal and spatial variations in POC export and distribution among different size fractions (Ran et al., 2013; Wang 2012; 2016; Yu et al., 2019a, b ; Qu et al., 2020); 3) impact of anthropogenic activities (Hu

et al., 2015; Tao et al., 2018; Yu et al., 2019a); and 4) burial efficiency and preservation in the ocean (Sun et al., 2018; Tao et al., 2016; Ge et al., 2020). However, all these previous studies rely on suspended sediment samples collected near the channel surface or at a single, intermediate depth in the river channel, further assuming a homogeneous distribution of suspended sediment characteristics in the water column, both vertically and laterally. It is now well recognized that suspended sediments present physical, mineralogical, chemical, and isotopic heterogeneities across river transect due to hydrodynamic sorting and tributary mixing (Galy et al., 2008b; Garzanti et al., 2010; Bouchez et al., 2010, 2011a). This is also true for POC, whose age and composition vary following vertical water depth (*e.g.*, Galy et al., 2008b; Bouchez et al., 2014; Hilton et al., 2015; Repasch et al., 2021; Schwab et al., 2022), and lateral river transect (*e.g.*, Bouchez et al., 2014; Baronas et al., 2020), and between sediment size fractions separated in the laboratory (Yu et al., 2019b; Ge et al., 2020). Such heterogeneity warrants a re-evaluation of POC transport in the Huanghe, accounting for the variability in suspended sediment characteristics over the channel cross-section.

In this study, we take advantage of in-river hydrodynamic sorting to access the full range of suspended sediment size fractions by collecting suspended particulate matter (SPM) samples along several river depth profiles distributed across a channel transect (*e.g.*, Bouchez et al., 2014; Freymond et al., 2018; Baronas et al., 2020). We apply this sampling scheme to a cross-section of the Huanghe located 200 km upstream from the river mouth and report SPM OC content, stable isotope composition, and radiocarbon activity as well as total nitrogen, major element composition (aluminum and silicon), and particle size distribution. Based on these novel samples and data sets, this study aims at 1) determining the controls on POC content in the Huanghe; 2) tracing and quantifying the sources of riverine POC in the Huanghe; and 3) providing depth-integrated estimates of POC fluxes in the most turbid large river system.

## 2 Study area

The Huanghe originates from the north-eastern Qinghai-Tibet Plateau (QTP) and runs through the Chinese Loess Plateau (CLP) and the North China Plain (NCP) to the Bohai Sea (Figure 1a). It is 5,464 km long and drains a basin area of $79.5 \times 10^4$ km². The Huanghe drainage basin can be subdivided into three main geomorphic units: 1) the high-relief upper reaches spanning from the source region (elevation of 4,500 m) to the city of Toudaoguai (located 3,472 km downstream at an elevation of 1,000 m); 2) the middle reaches with a channel length of 1,206 km, ending at Huayuankou (elevation of 110 m) draining landscapes characterized by relatively gentle slopes; and 3) the lower reaches where the river flows eastwards across a fluvial plain over a length of 786 km. These three sections drain 53.8%, 43.3%, and 2.9% of the whole Huanghe basin area, respectively (Wang et al., 2007; YRCC, 2016). Most second-order tributaries drain the CLP region and feed the main channel in the middle reaches, the Dawenhe River being the only tributary of the lower reaches, with negligible water and sediment supply due to upstream trapping in lakes and reservoirs. It is worth noting that more than 50% of the water discharge at the Huanghe's mouth comes from the QTP, whereas over 90% of the sediment load originates from the CLP (Wang et al., 2010, 2017; Pan et al., 2016). The CLP is thus the principal source area of sediment to the Huanghe (Shi and Shao, 2000; Guo et al., 2002; Wang and Fu et al., 2016).

The Huanghe drainage basin is mostly underlain by the North China craton, and is bounded by several mountain belts. The watershed encompasses 46% of sedimentary rock outcrops (mainly siliciclastic rocks with minor carbonates), and

about 45% of unconsolidated sediments (mainly Quaternary loess deposits). The remaining outcrops include
metamorphic, plutonic, and volcanic rocks formed from the Archean to the Tertiary (Figure S1). Although river
incision is strong in the QTP, a substantial part of the corresponding eroded material is not effectively transferred to
the lower reaches due to deposition in the CLP and the western Mu-Us desert, a situation that has prevailed since at
least the middle Pleistocene (Nie et al., 2015; Licht et al., 2016; Pan et al., 2016). In addition, recent anthropogenic
disturbance such as constructions of large dams in the upper reaches has profoundly modified the export of solid
materials from the basin (Wang et al., 2007). The Huanghe then flows through the CLP that has acted as the major
supplier of sediment to the system since at least the Calabrian Pleistocene (Stevens et al., 2013; Bird et al., 2015).
There, an easily erodible loess-paleosol formation has accumulated since 2.58 Ma (Guo et al., 2002), over a thickness
ranging from a few meters to more than 500 m, with an average of 100 m. This loess-paleosol formation and underlying
Cretaceous sedimentary rocks are actively incised by the main stem and its tributaries (Shi and Shao, 2000; Guo et
al., 2002; Wang and Fu et al., 2016). Notably, the Ordos Basin underlying the CLP is rich in oil and gas (Guo et al.,
2014). In the lower reaches, the river drains Quaternary fluvial deposits and sedimentary rocks.
The Huanghe drainage basin encompasses the entire arid and semi-arid region of northern China in the upper and
middle reaches, and is characterized by more humid climate conditions in the lower reaches. Annual average
precipitation (over the period 1950 - 2000) in the upper, middle and lower reaches regions is 368 mm, 530 mm, and
670 mm, respectively (Wang et al., 2007). As a result of the East Asian summer and winter monsoon circulations, the
rainy season (June to September) contributes 85% of the annual precipitation (Wang et al., 2007). During the rainy
season, frequent storm events lead to concentrated flows (relatively high discharge) in vulnerable gully-hill systems,
the dominant regional geomorphic landscape, and actively participates in soil erosion in the CLP (Shi and Shao, 2000;
He et al., 2004; Qu et al., 2020). The present-day (2002 to 2016) suspended sediment flux delivered by the Huanghe
to the sea is about 0.12 Gt/yr, which implies a decrease of nearly 90% in sediment export compared to the widely cited
estimate of 1.08 Gt/yr (average value between 1950 to 1980, Milliman and Farnsworth, 2011). This massive decrease
in sediment export mostly results from human perturbations, including soil conservation practices in the CLP and
retention in large reservoirs, rather than from climatic variations such as the decreasing precipitation observed in the
region over the last decades (Wang et al., 2007; Ran et al., 2013; Wang and Fu et al., 2016; Li et al., 2022). A scheme
for water and sediment regulation (WSR) has been implemented through the construction of the Xiaolangdi Reservoir
since 2002, aiming to mitigate water and sediment imbalances in the lower reaches. This regulation has resulted in a
modification of the flux of sediment delivered to the lower reaches and estuary, making the Huanghe a highly human-
regulated river system. However, no WSR was implemented in 2016, the year of our sampling campaign, suggesting
that the collected SPM samples are not significantly affected by retention in dams, and thus are representative of the
fluvial transport of terrestrial materials eroded from the CLP.

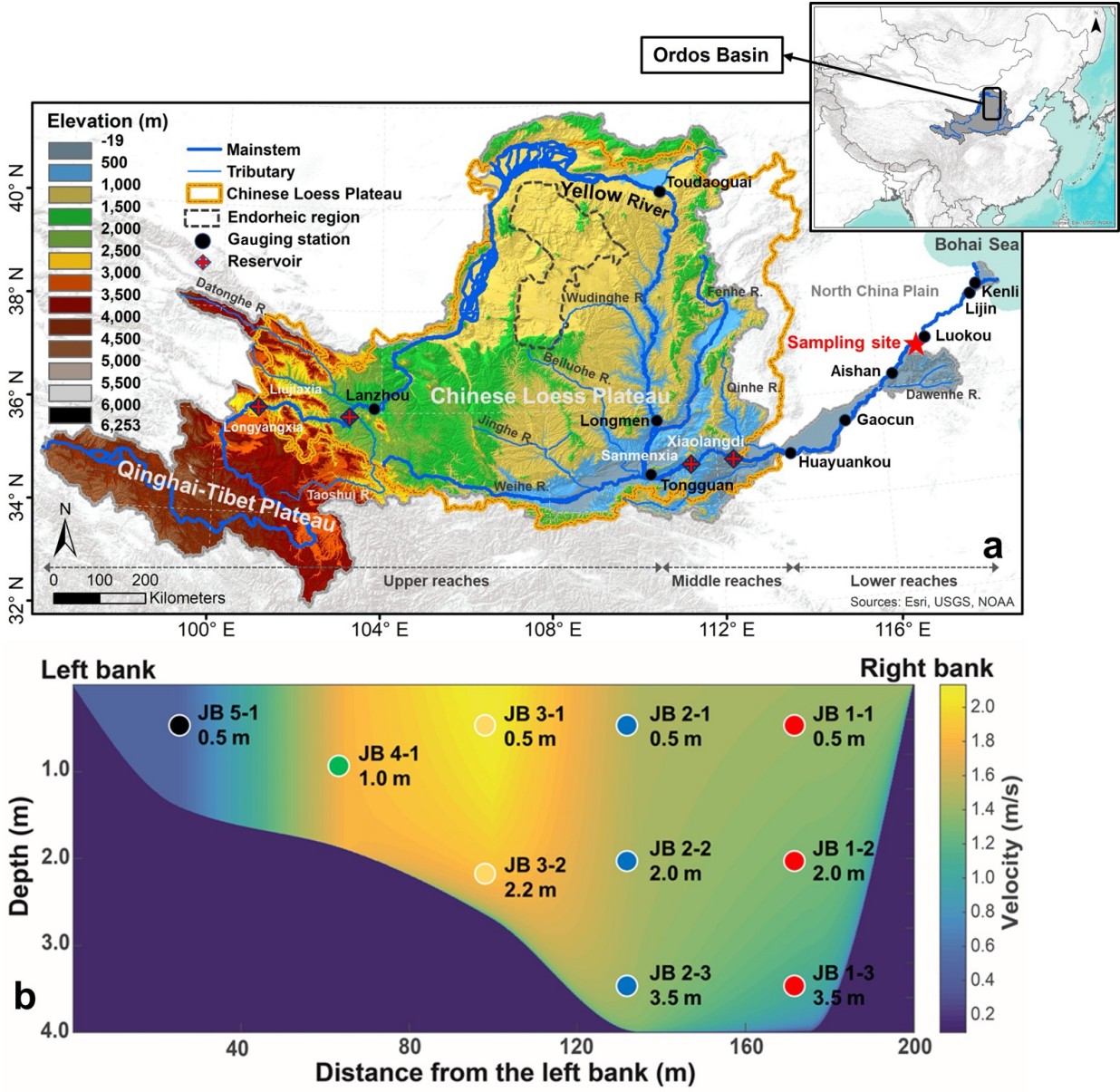

Figure 1: (a) Elevation map of the Huanghe drainage basin showing the main reservoirs and gauging stations along the main stem as well as our sampling site (36.75ºN, 117.02ºE, near the Luokou gauging station); and (b) channel cross-section sampled for this study showing the depth and lateral distribution of suspended particulate matter (SPM) samples and modeled velocity distribution based on the "law of the wall", using the point velocity data measured by a current velocity meter attached to the sampler.

## 3 Sampling and analytical methods

### 3.1 Sampling strategy

Detailed sampling of a cross-section of the Huanghe was carried out on the 17th of July, 2016, during the flood season (Figure S2). Samples were collected along five depth profiles near the Luokou hydrological station (36.75ºN, 117.02ºE), 250 km upstream from the river mouth (Figure 1). This sampling strategy allows for accessing the full range of suspended sediment particle size (Bouchez et al., 2014). The cross-section is 200-meters wide at the surface and 4

meters deep at most (Figure 1; YRCC, 2016). As in previous studies, we used a home-made, 10-liter, point-sediment
horizontal Niskin-type sampler attached to a current velocity meter, to collect river water samples and measure the
water velocity simultaneously. Subsequently, two water samples were collected at the surface near the right bank in
May and June 2017 before the flooding season, to retrieve fine suspended particulate matter. For each sample,
approximately 30 liters of river water were collected and were then filtered through pre-weighed 0.22-μm porosity
cellulose acetate membrane filters within 24 hours. After rinsing the filters with filtered water, all sediment samples
were transferred into centrifuge tubes and freeze-dried before weighing and analysis. A bed sediment (BS) sample
was collected on an exposed, recently flooded sediment bar of the riverbed.
**3.2 Physical and geochemical analysis**
Apart from a 50-mg aliquot of SPM samples preserved for particle size analysis, samples were finely ground using an
agate mortar and pestle prior to chemical and isotopic analyses. The particle size distribution of the unground aliquots
was measured using a Laser Diffraction Particle Size Analyzer (Beckman Coulter LS-12 320) at the École Normale
Supérieure (ENS), Paris, France. Before analysis, unground SPM aliquots were dispersed in deionized water and then
in sodium hexametaphosphate in an ultrasonic bath. For each sample, we measured three replicates and report the
average median particle size (D50, μm) with an uncertainty better than 2% (Table 1). The chemical composition of
SPM samples was measured on ground aliquots at the Centre de Recherches Pétrographiques et Géochimiques
(CRPG), Vandoeuvre-lès-Nancy, France, using inductively coupled plasma atomic emission spectroscopy (ICP-OES)
for major elements with typical uncertainties of 3% (Carignan et al., 2001).
For particulate organic carbon content (POC%, wt.), stable carbon isotope $\delta^{13}$C (in ‰$_{VPDB}$, *i.e.*, in ‰ relative to Vienna
Pee Dee Belemnite) and radiocarbon isotope $\Delta^{14}$C (expressed as fraction modern, Fm), ground homogenized samples
were fumigated using 12M HCl fumes in a closed Teflon tank at 60 °C for 48 hours to remove the carbonate fraction,
and were then dried under vacuum prior to analysis. Total nitrogen content (TN%, wt.) was measured on non-acidified
samples (Komada et al., 2008). Triplicate analysis on POC% and $\delta^{13}$C of POC (acidified aliquots) as well as TN%
(non-acidified aliquots) were carried out on an Organic Elemental Analyzer (OEA) coupled with Isotope Ratio Mass
Spectrometry (IRMS, Thermo Scientific Flash 2000) under continuous flow mode at Géosciences Paris Saclay
(GEOPS), Orsay, France. Subjected to the blank subtraction by linearity test, two international standards including
USGS-40 and IAEA-600 as well as an internal standard (GG-IPG) were used to build linear regression equations to
calibrate the elemental and isotopic values for both carbon and nitrogen. Uncertainties on POC%, $\delta^{13}$C, and TN%,
based on replicate measurements (1σ, n=3), are lower than 0.02%, 0.06‰, and 0.02‰, respectively. The $^{14}$C activity
of POC was measured on a new compact accelerator mass spectrometry (AMS), *ECHo*MICADAS (Hatté et al., 2023),
using a gas ion source interface system (GIS) at the Laboratoire des Sciences du Climat et de l'Environnement (LSCE),
Gif-sur-Yvette, France, with an absolute uncertainty of max ±0.5%. Aside from the gas bottles of prepared blank PhA
and standard NIST OX II which are permanently connected to the GIS and, used for normalization and corrections for
fractionation and background, international standards including IAEA-C5, IAEA-C7, IAEA-C8, and blank PhA were
prepared in different sizes (10 to 100's μg C) to match the amount of OC found in the sediment samples.

### 3.3 POC source apportionment

To quantify the contribution and associated uncertainties of various sources to POC transported in the Huanghe, a Bayesian Markov Chain Monte Carlo (MCMC) based on a three-end member (Appendix A) mixing scheme was adopted. This approach considers the variability on each end member contribution, assuming this variability can be represented by a normal distribution. We computed the *a posteriori* distribution of the Bayesian formulation using the MCMC method, using the MixSIAR package (Moore & Semmens, 2008; Stock & Semmens, 2016). All computations were performed in the R environment (http://www.r-project.org/). To ensure reliable simulation, the model was run with chain length of 300,000 by 3 chains, using a burn-in of 200,000 steps, and a data thinning of 100 for each sample. The mixing model was constructed on the dual stable and radioactive isotope of the riverine POC pool ($\delta^{13}C$ and $\Delta^{14}C$) and of the three potential source pools (Section 5.2) by the following equations:

$$\text{Isotope\_ratio}_{\text{sample}} = \sum_{\text{source}} (f_{\text{source}} * \text{Isotope\_ratio}_{\text{source}})$$

$$\sum_{\text{source}} f_{\text{source}} = 1$$

where $\text{Isotope\_ratio}_{\text{sample}}$ is either the $\delta^{13}C$ or $\Delta^{14}C$ value of the sample, $\text{Isotope\_ratio}_{\text{source}}$ is either the $\delta^{13}C$ or $\Delta^{14}C$ value of different possible sources of POC and $f_{\text{source}}$ is the relative contribution of each source of POC. Further model diagnostics was performed using Gelman-Rubin and Geweke test, both diagnostics validated the robustness and convergency of the model.

### 3.4 Depth-integrated fluxes

Instantaneous depth-integrated fluxes of SPM and POC sources were calculated for the cross-section using a method developed by Bouchez et al. (2011a, b). This method is based on the systematic variation of SPM concentration in the water column (Figure 2) applying a Rouse-based model (Rouse, 1937). We first constructed a bathymetric profile of the river cross-section based on the depth information collected in the field and then modeled the velocity distribution across the transect (Figure 1b) through fits of the so-called "law of the wall" to water velocity measured at the location of each sample within the cross-section using a current meter. Afterward, the concentration of total SPM and various particle size fractions could be estimated by applying the so-called Rouse model (Rouse, 1937) to each particle size fraction separately (Bouchez et al., 2011a), resulting in a map of the particle size distribution in the river cross-section (Figure S3). The aluminum to silicon ratio (Al/Si mass ratio) is inversely related to the particle size of river SPM in the Ganges-Brahmaputra, the Amazon, and the Mackenzie Rivers (Galy et al., 2007; Bouchez et al., 2014; Hilton et al., 2015). Such a linear relationship between D50 and Al/Si was also observed in our dataset, allowing for computing the spatial distribution of POC content in the cross-section, based on the linear relationship between POC and Al/Si (Figure 3). Finally, combining modeled water velocity, SPM concentration, and POC distribution we calculated a depth-integrated, instantaneous POC flux for the whole river channel (Figure S3, detail in supplementary material).

## 4 Results

We report the first isotopic dataset of POC samples collected along several depth profiles distributed over a cross-section of the Huanghe (Table 1, n=10). SPM concentrations range from 679 to 2,459 mg/L (avg. 1,286 ± 572 mg/L, reported with mean value with one standard deviation, herein after) and show an obvious increase from the surface to the bottom and from the right bank to the left bank (Figure 1b and 2a). The surface SPM concentration (*i.e.*, samples collected 0.5 m below the surface) decreases laterally as the water column deepens. The range of measured Huanghe SPM D50, *i.e.* the median particle size (19.5-86.0 μm, Figure 2b) agrees with that of SPM collected at Lijin (16.6-120.1 μm, n=50) during the same flooding season by Moodie et al. (2022). In each depth profile, SPM is consistently coarsening with depth as revealed by the evolution of grain size parameters such as D10, D50, and D90 (Table 1, Table S1, Figure 2 and Figure S4). The finest SPM is transported on the right bank and at the surface, while the coarsest SPM is found at the bottom of the middle profile (sample JB 2-3). Two types of depth profiles can be distinguished at Luokou based on particle size distributions (Figure S4) and the relationship between D50 and water depth (Figure 2b). On the one hand, the JB 1 and JB 4 profiles show a well-marked, bi-modal distribution of particle size (Figure S4) together with relatively low and consistent D50 (Figure 2b). On the other hand, the JB 2, JB 3, and JB 5 profiles show a more unimodal distribution of particle size (Figure S4) and a unique D50 - sampling depth relationship (Figure 2b). Interestingly, these two groups can also be distinguished in terms of relationships between POC% and $\delta^{13}C$ with water depth (Figures 2d, 2f). As expected, the Al/Si ratio is well-related to the particle size, and the ratios measured in the middle profile SPM samples (0.17 for JB 2-3 and 0.26 for JB 2-1) encompass the full range of Al/Si found in the whole cross-section (Figure 2c). The relatively low Al/Si ratios are comparable to that of the middle Huanghe (Qu et al., 2020) and other large turbid river systems such as the Ganges-Brahmaputra (Galy et al., 2008b), Salween, and Irrawaddy (Tipper et al., 2021).

SPM in the Huanghe is characterized by low TN and POC content (wt.%), ranging from 0.04% to 0.08% (0.06 ± 0.01%) and from 0.29% to 0.42% (0.37 ± 0.06%), respectively (Figure 2d; Table 1). POC content generally decreases from the surface to the river bed, with quantitative differences from one profile to another (Figure 2d). Notably, the JB 1 profile shows the highest POC% and TN%. In addition, the ratio of TN% to POC%: $N/C_{org}$ increases with depth in the JB 1 profile (from top to bottom), while it decreases in the JB 2 and JB 3 profiles (Figure 2e). The $\delta^{13}C$ of POC varies over a narrow range from −26.55‰ to −25.75‰ (−26.12 ± 0.29‰, Figure 2f) and becomes lighter with depth, showing that fine SPM has higher $\delta^{13}C$ than coarse SPM. These values are lower than those previously reported for other Huanghe sampling sites upstream: −24.7 ± 0.4‰ at Toudaoguai, −24.9 ± 0.6‰ at Longmen, and −23.8 ± 0.6‰ at Lijin (Qu et al., 2020, Hu et al., 2015; Tao et al., 2015; Yu et al., 2019a; Ge et al., 2020). The radiocarbon activity of POC of the Huanghe at Luokou is relatively low (Figure 2g), with Fm ranging from 0.552 ($\Delta^{14}C$ = −453‰; sample JB2-3) to 0.675 ($\Delta^{14}C$ = −331‰; sample JB1-3), spanning from 3,160 to 4,780 $^{14}C$ yrs, and the average value is 0.607 ± 0.038 ($\Delta^{14}C$ = −412 ± 37.6 ‰, n=10). This range of radiocarbon activity is consistent with published values for POC collected at the river surface downstream of Toudaoguai (Qu et al., 2020). All the POC radiocarbon activity data reported so far for the Huanghe are comparable to mean values for Arctic large rivers ($\Delta^{14}C$ = −397‰, ca. 4,480 $^{14}C$ yrs, Ke et al., 2022), revealing the multimillennial-age nature of POC transported by the Huanghe. The elemental and isotopic signatures of the two fine SPM samples HH 17.05 and HH 17.06 (on average POC% = 1.07%, $\delta^{13}C$ =

−25.67‰, $F_m$ = 0.720; and Al/Si = 0.37) are significantly different from those of the depth profile samples (Table 1).
The bed sediment sample has a comparatively low POC% (0.21%), $\delta^{13}C$ (−27.35‰), $F_m$ (0.099), and Al/Si ratio

250  (0.17).

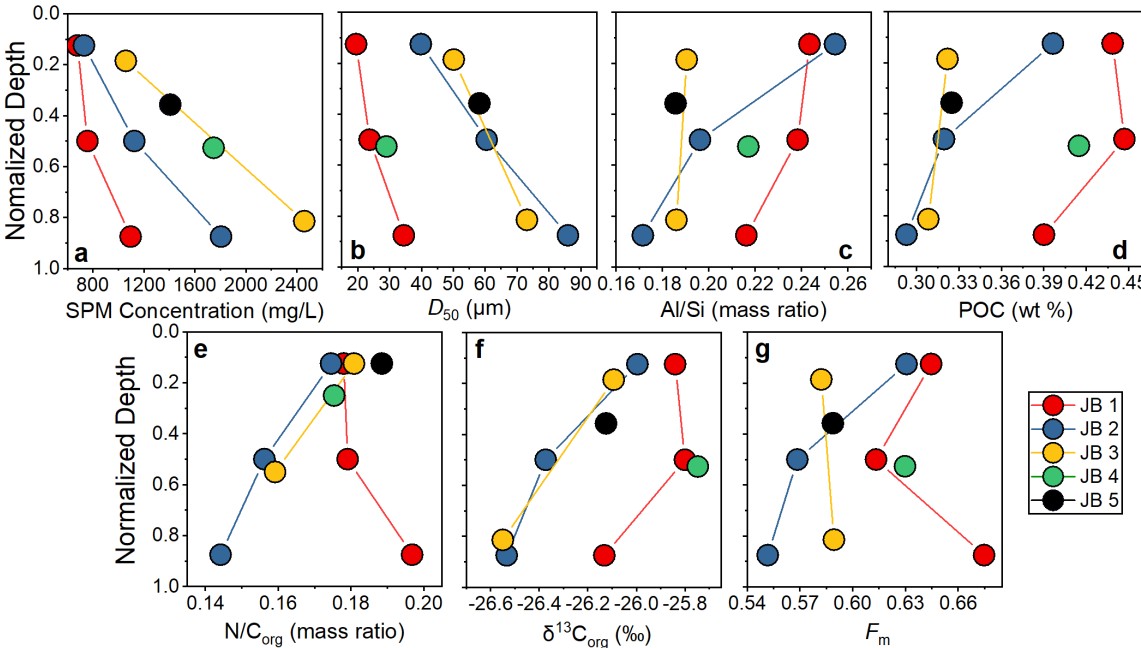

**Figure 2: Variation of physical and chemical parameters in the river cross-section, shown as a function of sampling depth**
**normalized to total depth of the water column at the location of the considered depth-profile, for the Luokou cross-section**
**on the Huanghe (June 16, 2017). (a) SPM concentration; (b) particle size distribution (shown as D50); (c) Al/Si mass ratio;**
**(d) POC content (weight %); (e) N/$C_{org}$ mass ratio; (f) stable carbon isotope ratio $\delta^{13}C_{org}$ (‰); (g) radiocarbon activity $F_m$.**
**Table 1: SPM characteristics and POC properties of the river-cross-section sampling.**

| Sample ID | Type | Depth (m) | SPM mg/L | POC (%) | SD | $\delta^{13}C_{org}$ (‰) | SD | $F_m$ | $\Delta^{14}C$ (‰) | SD | $^{14}C$ age | TN (%) | N/$C_{org}$ | Al/Si | D50 μm |
|---|---|---|---|---|---|---|---|---|---|---|---|---|---|---|---|
| JB 1-1 | SPM | 0.5 | 679 | 0.44 | 0.02 | −25.84 | 0.03 | 0.645 | −360 | 7 | 3527 | 0.078 | 0.178 | 0.243 | 19.5 |
| JB 1-2 | SPM | 2 | 757 | 0.45 | 0.01 | −25.80 | 0.06 | 0.613 | −392 | 11 | 3929 | 0.080 | 0.179 | 0.238 | 23.7 |
| JB 1-3 | SPM | 3.5 | 1095 | 0.39 | 0.01 | −26.13 | 0.04 | 0.675 | −331 | 7 | 3161 | 0.077 | 0.197 | 0.216 | 34.5 |
| JB 2-1 | SPM | 0.5 | 730 | 0.40 | 0.03 | −26.00 | 0.01 | 0.631 | −374 | 7 | 3703 | 0.069 | 0.175 | 0.255 | 39.8 |
| JB 2-2 | SPM | 2 | 1124 | 0.32 | 0.01 | −26.37 | 0.06 | 0.569 | −436 | 13 | 4537 | 0.050 | 0.156 | 0.196 | 60.4 |
| JB 2-3 | SPM | 3.5 | 1806 | 0.29 | 0.01 | −26.53 | 0.06 | 0.552 | −453 | 38 | 4779 | 0.042 | 0.144 | 0.172 | 86.0 |
| JB 3-1 | SPM | 0.5 | 1058 | 0.32 | 0.02 | −26.09 | 0.03 | 0.582 | −422 | 12 | 4346 | 0.058 | 0.181 | 0.190 | 50.1 |
| JB 3-2 | SPM | 2.2 | 2459 | 0.31 | 0.02 | −26.55 | 0.04 | 0.589 | −415 | 11 | 4247 | 0.049 | 0.159 | 0.186 | 73.1 |
| JB 4-1 | SPM | 1 | 1747 | 0.41 | 0.02 | −25.75 | 0.06 | 0.630 | −375 | 8 | 3714 | 0.073 | 0.175 | 0.217 | 29.0 |
| JB 5-1 | SPM | 0.5 | 1406 | 0.32 | 0.01 | −26.12 | 0.05 | 0.589 | −416 | 11 | 4256 | 0.061 | 0.188 | 0.186 | 58.2 |
| HH 17.05 | SPM | 0 | 83 | 0.92 | 0.00 | −25.73 | 0.14 | 0.711 | −295 | 19 | 2740 | 0.184 | 0.200 | 0.358 | 5.2 |
| HH 17.06 | SPM | 0 | 54 | 1.21 | 0.01 | −25.60 | 0.07 | 0.729 | −277 | 25 | 2539 | 0.261 | 0.215 | 0.377 | 4.3 |
| HH | BS | | | 0.21 | 0.03 | −27.35 | 0.05 | 0.099 | −901 | 7 | 18539 | 0.019 | 0.087 | 0.175 | 44.4 |

**5 Discussion**
We observe significant heterogeneities of elemental and isotopic carbon composition as well as inorganic chemistry
over the studied river cross-section. The possible mechanisms behind these variations are assessed hereafter. Then,
sources of riverine POC are determined and quantified, confirming that erosion of the loess-paleosol sequence of the
CLP is a major source of aged and refractory biospheric OC to the Huanghe. Finally, we assess the POC load and its
variability over the transect profile, inferring the importance of the supply of POC from the river bottom in the
Huanghe.

**5.1 Transportation mode of POC in the Huanghe**

**5.1.1 POC loading and its controls**

The Huanghe is characterized by a high SPM load with relatively low POC% ($0.37 \pm 0.06\%$, n=10). In the Luokou
cross-section, POC content generally increases with decreasing particle size (Figure S5), with the two clay-sized
("HH") samples showing the largest POC content (Table 1). Consistently, the Al/Si ratio of Huanghe sediments, which
varies as an inverse linear function of the median particle sizes D50 ($R^2=0.72$, $p >.001$, Figure 3a), positively correlates
with POC% ($R^2=0.81$, $p <.001$ Figure 3b), a pattern observed globally (Galy et al., 2008b; Bouchez et al., 2014; Hilton
et al., 2015; Repasch et al., 2021). This pattern is consistent with POC variability in the Huanghe reported for manually
separated size fractions of sediments (Yu et al., 2019b).

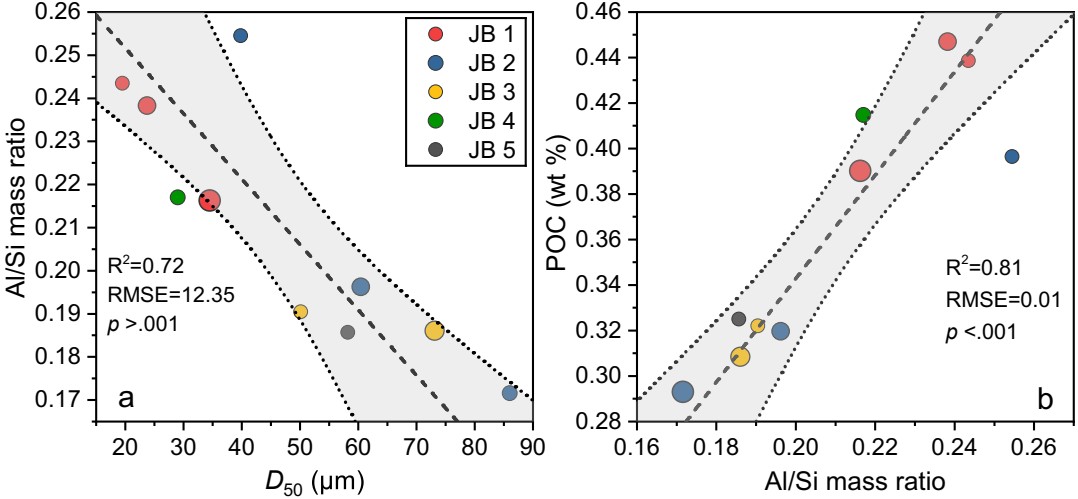

**Figure 3: Relationships between (a) particle size D50 and Al/Si mass ratio; (b) Al/Si mass ratio and POC content for the**
**Luokou cross-section on the Huanghe (July 17, 2016). The symbol size indicates the sampling depth in the water column,**
**with symbol size increasing with depth. The shaded area represents the 95% confidence area of the linear best-fit (black**
**dashed line), the upper and lower bound are marked by grey dotted lines.**
The first reason for the low POC content of Huanghe sediments is therefore their relatively low values of Al/Si
compared to other systems - a feature than can be related to the quartz-rich, OC-poor nature of the loess-paleosol
formations of the CLP (Jahn et al., 2021; Huang and Ren, 2006; He et al., 2006; Ning et al., 2006; Wang and Fu,
2016). However, Huanghe sediments are relatively poor in POC, even considering their low Al/Si, compared to other
rivers globally. To that effect, the so-called "POC loading" can be characterized by the slope described by sediment
data in an Al/Si-POC diagram (Galy et al., 2008b; Figure 4). For a given Al/Si ratio, the POC% in the Luokou cross-
section is similar to that of the middle Huanghe (Qu et al., 2020), indicating the relatively invariant transport mode of
POC between the middle and lower reaches. Previous studies have shown that the positive relationship between POC%
and Al/Si can be partially explained by OC adsorption onto the mineral surface (Curry et al., 2007; Galy et al., 2008b;
Blair and Aller, 2012; Bouchez et al., 2014; Qu et al., 2020). In the loess-paleosol deposits acting as a source of
sediments to the Huanghe, OC is mostly preserved and stabilized by forming organo-aggregates with kaolinite and
through adsorption onto iron oxides (Wang et al., 2013).
However, POC loading in the Huanghe is small compared to that of the Amazon (Bouchez et al., 2014), but similar
to that of the Ganges-Brahmaputra system (Galy et al., 2008b). While many factors could influence POC loading
across these catchments, we note that another similarity between the Huanghe and Ganges-Brahmaputra fluvial
systems is their millennial-aged $OC_{bio}$ (Galy et al., 2007; Tao et al., 2016). This is in stark contrast with the Amazon,
where younger $OC_{bio}$ ages have been reported (Bouchez et al., 2014). Given that younger $OC_{bio}$ recently
photosynthesized in terrestrial or aquatic ecosystems can be readily oxidized within catchments (Mayorga et al., 2005),
the relatively low POC loading observed at the mouth of the Ganges-Brahmaputra and the Huanghe (Figure 4) could
be related to the predominance of refractory, aged $OC_{bio}$ and $OC_{petro}$ in those systems, while the Amazon sediments
would still contain significant amount of younger, more labile $OC_{bio}$.

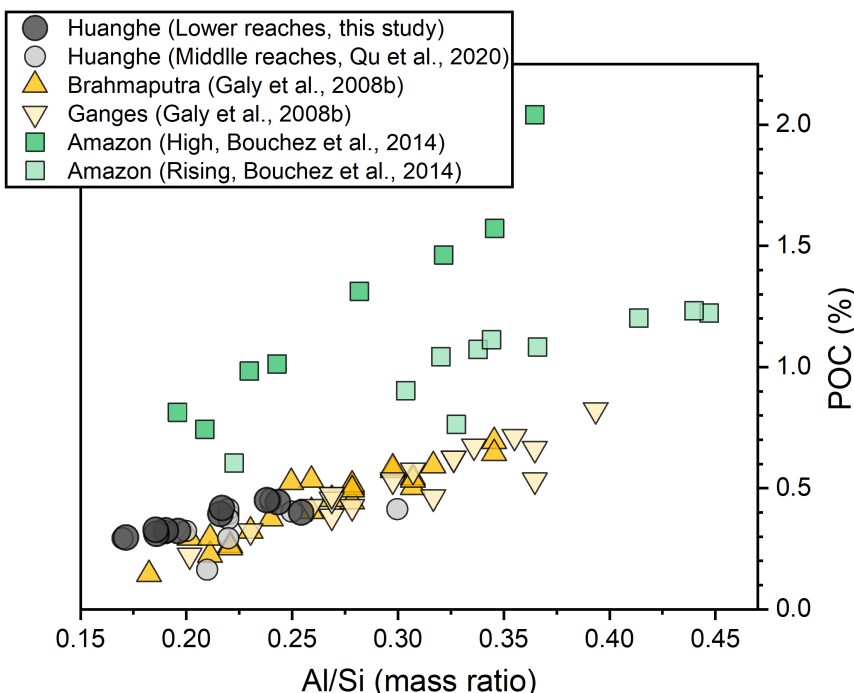

**Figure 4: "POC loading" of river SPM of large rivers. The POC loading is estimated from the slope of the relationship**
**between POC content and the Al/Si ratio of each fluvial system (Galy et al., 2008b). All SPM samples were collected along**
**depth profiles except for the middle Huanghe (Qu et al., 2020).**
In detail, and as explained in more detail below (**Section 5.1.2**) we also observe a significantly different POC loading
between the JB 1 and JB 2 depth profiles at the Luokou station (Figure 3). This difference in POC loading in the cross-
section of the Huanghe might indicate the delivery of recent $OC_{bio}$, specifically near the right bank (the closest to the
JB 1 profile) for the Luokou site, a scenario which is supported by the comparatively younger age of POC in profile
JB 1. Consistently with this interpretation, temporally variable POC loading at a given site has been reported for the
Amazon (Bouchez et al., 2014), where higher POC loading during the high-water stage compared to the rising water
stage has been attributed to the erosion of discrete organic debris from riverbanks.
Variable POC loading amongst large catchments has implications for evaluating the likelihood of POC preservation
in estuaries. The Ganges-Brahmaputra system delivers relatively old, refractory $OC_{bio}$ to the Bengal Fan with an almost
complete burial efficiency (Galy et al., 2007). Given the observed similarity in POC loading and age, we can thus
expect a similar, efficient preservation for the Huanghe offshore depositional system. In addition to the low reactivity
of the POC transported by the Huanghe, the high sediment accumulation rates in the Huanghe coastal domain might
further inhibit OC oxidation (Blair and Aller, 2012). Consequently, the case of the Huanghe differs drastically from
that of the Amazon, where higher POC loading is observed, with a larger contribution of young, labile $OC_{bio}$ either as
discrete organic matter or associated with mineral surfaces, leading to low POC burial efficiency in the ocean
(Bouchez et al., 2014; Blair and Aller, 2012).

### 319 5.1.2 Chemical heterogeneity within the transect

There is clear lateral and vertical variability of POC content and SPM inorganic chemistry across the Luokou cross-
section of the Huanghe. For each vertical depth profile, clay-rich fine particles are transported near the channel surface,
and quartz-rich coarse particles flow near the river bottom. Accordingly, the Al/Si ratio, POC content and POC
radiocarbon activity generally decrease with depth. Elemental (POC%) and isotopic POC signatures ([13]C and [14]C) are
inversely related to the particle size (D50; Figure S5). These patterns are observed in other large fluvial systems, e.g.,
Ganges and Brahmaputra, Amazon and Mackenzie (Galy et al., 2008b; Bouchez et al., 2014, Hilton et al., 2015),
showing that hydrodynamic sorting is the primary control on suspended sediment OC content, segregating inorganic
and organic material according to particle size (Bouchez et al., 2011a, 2014).
At the Luokou sampling site, lateral variability at the channel surface shows that POC-rich fine particles are
preferentially transported near the right bank (Figure 2 and Figure S3). This pattern is validated by the Rouse model
provided in Text S1, the Rouse number ($Z_R$) is 0.137, 0.236, and 0.284 for JB-1, JB-2, and JB-3, respectively. In
essence, $Z_R$ can reflect the balance between gravitation settling and upward turbulent diffusion. $Z_R$ is smaller near the
right bank while larger near the left bank, showing heterogeneity across the transect. Larger particles exhibit a faster
settling velocity due to their increased weight, leading to a higher $Z_R$. On the other hand, the lighter ones settle more
slowly, resulting in $Z_R$ approaching 0. This means that their concentration remains relatively constant along a given
depth profile. However, as depth increases and the concentration of larger particles grows, the proportion of these
finer particles in the overall sediment decreases (Bouchez et al., 2011a). The channel geometry thus needs to be
examined as a potential factor to produce such lateral heterogeneity, in particular the mechanisms of bed sediment
resuspension and bank erosion.
Resuspension of bed sediments is also a possible mechanism that could explain the lateral heterogeneity in POC
content in the study cross-section of the Huanghe. Indeed, scouring of channel bed sediment at high water flow may
also shift POC to more negative radiocarbon and stable isotope signatures. Our sample set collected in July 2016
during a flooding period (water flow velocity up to 2.1 m/s, Figure 1) supports this scenario. Indeed, the increase in
D50 of surface SPM samples from right to the left bank, that is with total channel depth decrease, is consistent with
coarse sediment resuspension from the bed. This is also supported by the Rouse model, where higher $Z_R$ in the shallow
water near the left bank indicates a greater likelihood of sediment settling to the bed, lower $Z_R$ suggests that there is
enhanced SPM supply from the riverbed. Such a scenario is also supported by the three-fold increase in SPM flux
observed from the upstream Huayuankou station to the downstream Lijin station in July 2016, despite a four-fold
decrease in water discharge (Figure S2).
Bank erosion can be a significant mechanism for the delivery of sediments to river systems (Guo et al, 2007). Bank
erosion at Luokou would make OC from the lower Huanghe alluvial plain a potential source of POC in the lower
reaches of the Huanghe. Frequent inundation to the adjacent riparian zones in flooding seasons, surface runoff driven
by storm events, and agriculture irrigation etc., can mobilize young soil OC and discrete organic matter debris (*e.g.,*
plant-drived debris) to riverine POC (Hilton et al., 2011; Turowski et al., 2016). This mechanism provides a possible
explanation for the opposite trends displayed by samples from the JB 1 and JB 2 profiles in the Fm *vs.* $\delta^{13}$C space
(Figure 5). The youngest POC was found at the bottom of the JB 1 profile (JB 1-3). Meanwhile, the JB 1 samples have
comparatively higher $N/C_{org}$ ratios and N%, consistent with the input of discrete plant-derived debris from the bank
in addition to rock-derived detrital clastic material in the coarse fractions (> 32 μm, Yu et al., 2019b). The transport
and entrainment of plant debris deep in the water column has been evidenced in many large river systems, such as the
Amazon (Feng et al., 2016), the Ganges-Brahmaputra (Lee et al., 2019), the Mackenzie (Schwab et al., 2022), the Rio
Bermejo River (Repasch et al., 2021). Such input would also provide an explanation for the higher POC loading of
the JB 1 profile (**Section 5.1.1**).

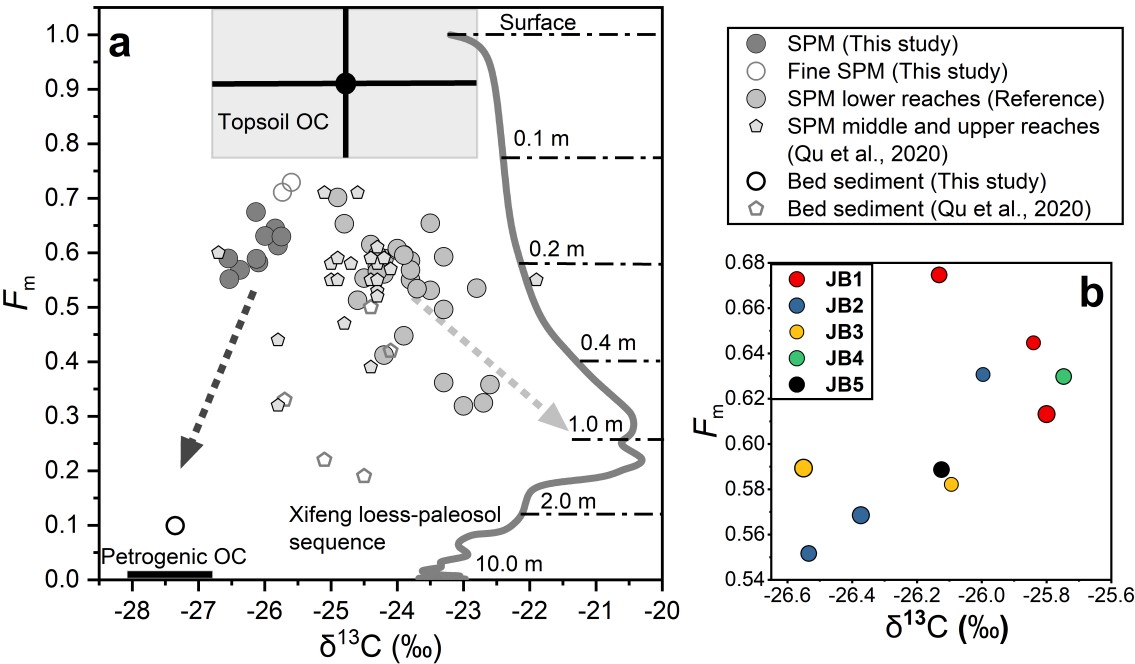

**Figure 5: (a) $^{14}$C activity (expressed as Fm) *vs.* $\delta^{13}$C for a compilation of POC data collected over the 2011-2016 period in**
**the lower Huanghe, including samples from this study and previous studies at Huayuankou, Lijin, and Kenli (Hu et al.,**
**2015; Tao et al., 2015; Yu et al., 2019a; and Ge et al., 2020); SPM and bed sediment (BS) collected by Qu et al., 2020 at**
**Toudaoguai (most downstream location of the upper reaches) and Longmen in the middle reaches (Table S1). The grey**
**curve corresponds to $\delta^{13}C_{org}$ of the top 10 m of the Xifeng loess-paleosol (Ning et al., 2006), and the corresponding Fm was**
**calculated from $^{10}$Be-derived ages following 'Age = −8033*ln(Fm)'. The soil depth is marked above the dot-dash line (Zhou**
**et al., 2010). Topsoil OC represents OC from the upper 10 cm of the loess-paleosol sequence with standard deviation marked**
**with black lines. (b) $^{14}$C activity (expressed as Fm) *vs.* δ$^{13}$C diagram for the Huanghe sediment samples collected in this**
**study at the Luokou cross-section. Symbol size increases with sampling depth in the water column.**

**5.2 POC provenance in the Huanghe: the significance of loess-paleosol-derived OC**

**5.2.1 Physical erosion of the loess-paleosol sequence**

Over decennial to centennial time scales, the POC export of the Huanghe is mainly controlled by erosion of the CLP.
Throughout the Quaternary, the erosion rate in the Huanghe basin has been mainly driven by climate shifts until human
activities started and profoundly impacted sediment fluxes in the mid-Holocene (He et al., 2006). The Huanghe has
experienced a 90% decrease in annual sediment load since the 1950s (Wang et al., 2015), caused by weakened soil
erosion to the CLP and sediment retention by dams (Wang et al., 2007; Ran et al., 2013; Wang and Fu et al., 2016; Li
et al., 2022). To determine the contributions of the various terrestrial OC components to Huanghe POC, we compiled
published POC carbon isotope data for sediments collected in the lower reaches from 2011 to 2016, after the
Xiaolangdi Reservoir was operated (Figures 5 and 6). This dataset shows that the radiocarbon ages of Huanghe POC
are considerably old (5,100 ± 1,700 $^{14}$C yrs, n=29), with a minor fraction of modern photosynthesized OC$_{bio}$ (Tao et
al., 2015; Yu et al., 2019a, b). This relatively $^{14}$C-depleted POC suggests the significant contribution of OC originated
from deep soil horizons within the catchment. Given that loess is easily erodible and that there is widespread gully
erosion in the catchment, more intensive erosion of the CLP can mobilize more soils as well as older OC from deep
soil horizons to fluvial transport. Therefore, higher sediment load in the river can be characterized by radiocarbon-
depleted POC. This is evidenced by the negative trend between $^{13}$C and Fm of POC for sediment samples collected in
the Huanghe over the 2011-2016 period (Figure 5a), suggesting that deep horizons of the loess-paleosol formations
are a plausible source for the $^{14}$C-depleted end member. Besides, the preferential erosion of bomb carbon affected,
recently photosynthesized and possibly degraded OC$_{bio}$ from the overlying topsoils (< 10 cm) most likely contributes
to riverine POC (Tao et al., 2015).
As such, variable contribution of aged and radiocarbon-free OC from deep horizons of loess-paleosol formations of
the CLP should have a significant impact on the elemental and isotopic signature of POC in the lower Huanghe.
Erosion of loess-paleosol can also explain the decreasing POC% with increasing SPM concentration at different sites
of the main channel (Ran et al., 2013; Qu et al., 2020), the negative relationship between SPM concentration and
corresponding POC Fm at Luokou (R$^2$=0.49, *p* <.001, Figure 6), and the low POC loading of the Huanghe (Section
5.1.1), as the deep horizons of the loess-paleosol sequences are OC-poor, and mostly host OC that is highly degraded
and refractory (Liu et al., 2012; Wang et al., 2013; Cheng et al., 2020). However, the slight increase of POC δ$^{13}$C with
increasing SPM concentration (R$^2$=0.29, *p* >.001) might indicate a significant supply of soil OC from loess-paleosol
shallower depth, as inferred from the δ$^{13}$C variation within the Xifeng loess-paleosol sequence (Figure 5a).

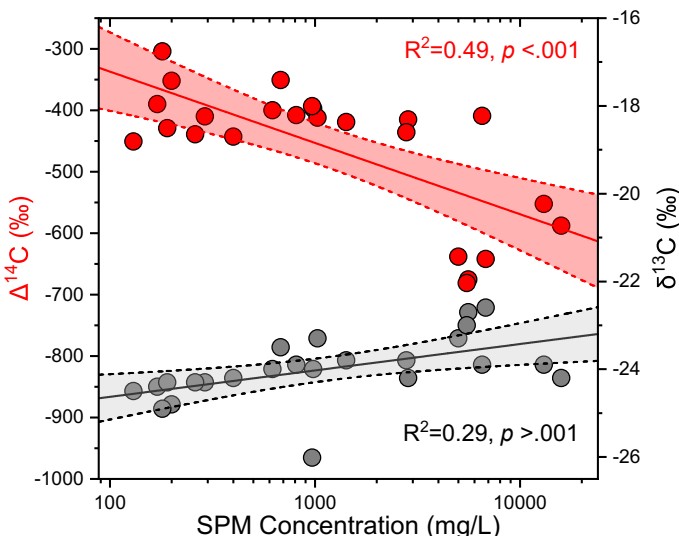

**Figure 6:** **$^{14}$C activity (expressed as Fm; red circles) and δ$^{13}$C (grey circles) of POC *vs*. SPM concentration for surface**
**samples from the Huanghe collected from 2011 to 2016 (average SPM concentration of surface samples in this study and**
**from Hu et al. (2015), Tao et al. (2015), Yu et al. (2019a), and Ge et al. (2020)). These paired dual carbon isotope data**
**corresponds to the group 'SPM lower reaches' in Figure 5. Straight lines correspond to best-fit logarithmic curves, and**
**shaded areas represent the 95% confidence interval.**

The N/C$_{org}$ ratio provides additional evidence for the significant contribution of loess-paleosol material to Huanghe POC (Figure S6). Indeed, the N/C$_{org}$ ratios of SPM collected in the lower reaches ranges from 0.10 to 0.23 (this study, Ran et al., 2013; Yu et al., 2019a), whereas topsoils of the CLP are characterized by N/C$_{org}$ lower than 0.14 (Liu and Liu, 2017) and sedimentary rocks typically have very low N/C$_{org}$ (Hilton et al., 2015). Soil OC input from the North China Plain is also unlikely given its N/C$_{org}$ of 0.10-0.13 (Shi et al., 2017). Therefore, all these possible sources cannot explain the high N/C$_{org}$ signatures of riverine SPM. In addition, the high turbidity of the Huanghe (> 600 mg/L) during the sampling season is likely to inhibit *in-situ* primary production (N/C$_{org}$ > 0.13) (Zhang et al., 2013; Hu et al., 2015). As a result, only soil OC from deep loess-paleosol horizons appears as a plausible supplier to downstream Huanghe POC, given the high N/C$_{org}$ ratios previously reported for various loess-paleosol sequences (Figure S6, Ning et al., 2006).

Geomorphic processes in the CLP region support the erosion of deep soil horizons. There, gully erosion is thought to be responsible for more than 80% of the total sediment yield in the CLP (He et al., 2006; Li et al., 2022). Gullies are densely distributed and cover about 42% of the total area of the CLP and up to 60% in hilly regions (Huang and Ren, 2006; He et al., 2006). Nowadays, the well-developed gully geomorphic system of the CLP is characterized by gullies with a depth of about 10 m on average and represents the most active vertical and regressive erosion of loess (Huang and Ren, 2006). This incision process erodes all types of unconsolidated materials, including the loess-paleosol sequence, underlaying red clays, and colluvial deposits in the form of creeps, falls, and slides in the watershed (Zhu, 2012). From 1925 to 1981, the erosion rate of the CLP was 6,318 t km$^{-2}$ yr$^{-1}$, compared to 10,770 t km$^{-2}$ yr$^{-1}$ in the hilly and gully plateau (Li et al., 2022). While the CLP's erosion rate dropped to 3,476 t km$^{-2}$ yr$^{-1}$ between 1982 and 2016, the rate in the hilly and gully plateau remained significantly high at 6,146.5 t km$^{-2}$ yr$^{-1}$ (Li et al., 2022). All these

observations suggest that gully erosion can strongly impact the composition of riverine POC. As gully erosion is
sensitive to climate change and anthropogenic activities, soil dynamics in the Huanghe basin have been altered since
the mid-Holocene (He et al., 2006; Li et al., 2022). Notably, the strengthening of the East Asian Monsoon in coming
decades (Li et al., 2022; Xue et al., 2023) could potentially enhance this process. However, in recent years, soil and
water conservation and environmental rehabilitation campaigns (Wang et al., 2007) largely contributed to the
reduction of SPM export by the Huanghe with a transfer to the estuary of 10.6 Mt in 2016, which is one order of
magnitude lower than the annual sediment flux measured in 2013 (172.8 Mt) and two orders or magnitude lower
compared to the flux of the 1950s (*ca*. 1,340 Mt; Wang et al., 2015). This sediment load reduction is consistent with
the weakened erosion rate observed in the CLP, such modifications should thus have drastically inhibited the OC
mobilization from the CLP and the POC export by the Huanghe.

### 5.2.2 POC source determination and end member apportionment

Considering the SPM geochemistry and the basin characteristics, three terrestrial sources can be identified as necessary
to form the composition of the Huanghe POC at the Luokou cross-section. As discussed in Section 5.2.1 and shown
in Figure 5a, two of these sources are (a) topsoil-derived OC ($OC_{ts}$) and (b) OC from deeper horizons of the loess-
paleosol sequence ($OC_{lps}$) excluding topsoil. In addition, at the Luokou cross-section, bed OC shows lower Fm and
$\delta^{13}$C values compared to that of SPM, suggesting a significant contribution of (c) rock-derived OC from erosion in the
middle reaches ($OC_{petro}$).
We adopted a Bayesian Monte-Carlo model to reconstruct source apportionment based on the mass balance of carbon
isotopes ($\delta^{13}$C and $\Delta^{14}$C) of our three defined end members (Section 3.3, Table. 2, Appendix A). The mixing space
showing the geometric area between three end members is shown in Figure S7. Modeling results are shown in Figure
7 as relative contributions (Figure 7a) and weight percentage (Figure 7b) of $OC_{ts}$, $OC_{lps}$, and $OC_{petro}$ (Table S2). The
contribution of $OC_{petro}$ to total Huanghe POC at Luokou varies between 10.1% and 13.9% in the cross-section, which
is higher than the contribution calculated for in the two fine SPM samples (avg. 9.1 ± 0.3%) and much smaller than
for the bed sediment sample (72.2 ± 10%). The inferred $OC_{petro}$ concentration in the sediment is remarkably uniform
in the cross-section, representing 0.04% of SPM (Figure 7b). This result is consistent with the OC contents of
midstream sedimentary rocks at 0.09 ± 0.08% (Qu et al., 2020). In addition, these findings imply that $OC_{petro}$
concentration does not depend on particle size and confirm previous findings of $OC_{petro}$ being present in a range of
clastic particles or as discrete particles (Galy et al., 2008a; Bouchez et al., 2014). In other words, the rock-derived OC
has a relatively invariant contribution with depth (Galy et al., 2008a; Bouchez et al., 2014), meaning that biospheric
OC exerts a first-order control on POC content and isotopic variations throughout the cross-section.
**Table 2: Summary of $\delta^{13}$C and $\Delta^{14}$C of source end members for POC in the Huanghe.**

| End member | $\delta^{13}$C | $\Delta^{14}$C |
|---|---|---|
| $OC_{ts}$ | −24.8 ± 1.9‰ | −90 ± 130‰ |
| $OC_{lps}$ | −22.7 ± 1.0‰ | −610 ± 390‰ |
| $OC_{petro}$ | −28.1 ± 1.5‰ | -1000‰ |

At the study cross-section, $OC_{ts}$ and $OC_{lps}$ contribute 46.6%-55.4% and 34.5%-39.5% to the total POC, respectively
(Figure 7a). The sum of these two components can be considered as $OC_{bio}$, which is 88.0 ± 1.3%. The corresponding
$OC_{bio}$ content of sediment is quite variable, ranging from 0.25% (sample JB 2-3) to 0.40% (sample JB 1-2), and
generally decreases from the river surface to the bottom. Given the rather invariant $OC_{petro}$ concentration in the
sediment, there are thus marked heterogeneities of POC provenance in the cross-section. For instance, POC
transported close to the right bank and in the finer SPM samples show a higher contribution from $OC_{bio}$. From the
knowledge of the relative contributions of $OC_{ts}$ and $OC_{lps}$ and their corresponding $^{14}C$ activity, the Fm values for the
bulk $OC_{bio}$ can be estimated based on mass balance. The modeled radiocarbon activity of $OC_{bio}$ varies from 0.64 to
0.75, corresponding to 3,570 to 2,300 $^{14}C$ yrs. In summary, our results support the first-order control of $OC_{bio}$
abundance on POC content and age in the Huanghe.

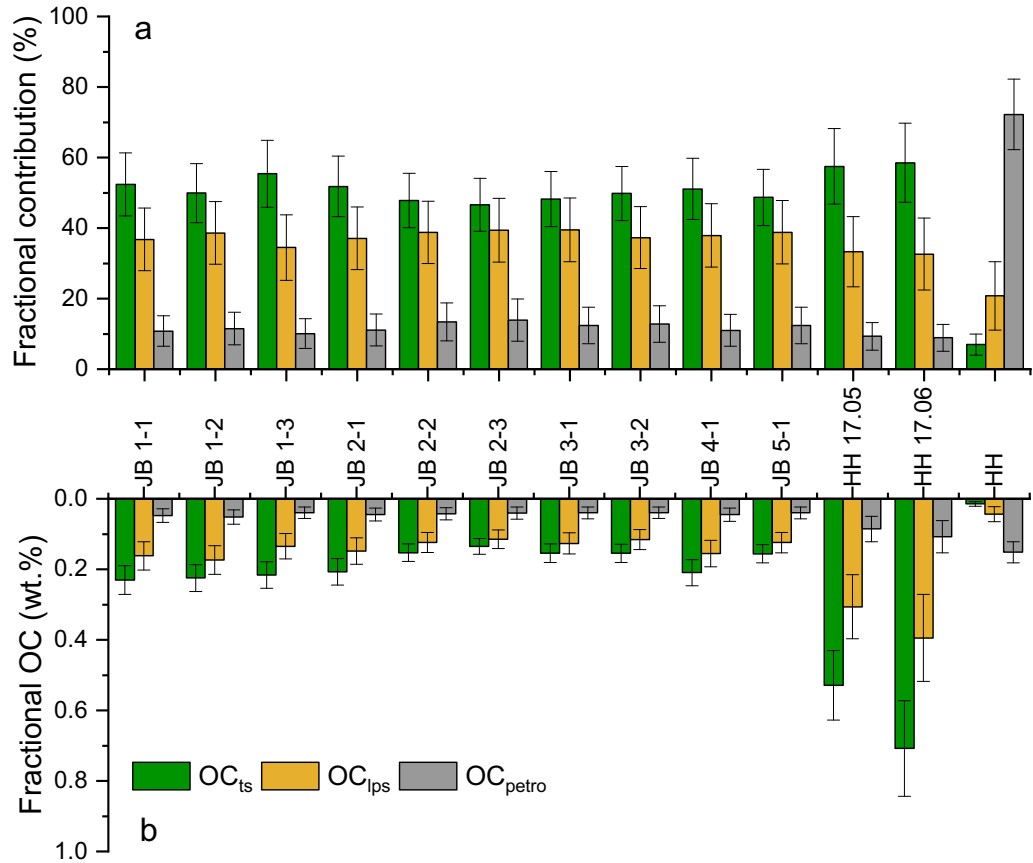

**Figure 7: (a) Relative contributions of the three different sources of Huanghe POC, (b) fractional OC weight percentage**
**Huanghe POC at the Luokou cross-section, as inferred from a mixing model. $OC_{ts}$ is the topsoil-derived OC, $OC_{lps}$**
**represents the loess-paleosol sequence OC excluding topsoil, and $OC_{petro}$ is the rock-derived OC eroded from the Huanghe**
**middle reaches.**
Applying the same mixing model to previously published Huanghe POC data (2011-2016, Table S1) shows (1)
dominance of the $OC_{bio}$ contribution to POC, (2) variable relative mixing proportions of $OC_{ts}$ and $OC_{lps}$; (3) a wide
range of $^{14}C$ age for $OC_{bio}$ (from 1,779 to 8,325 yrs). In particular, $OC_{ts}$ and $OC_{lps}$ contributed 28%-35% and 53%-
63% to POC collected in 2013 (Hu et al., 2015), leading to 75-89% of $OC_{bio}$. Yu et al. (2019a) estimated that $OC_{bio}$
contributed 63%-81% to the lower Huanghe POC (2015-2016) using a different mixing model. Using their data in our
mixing model results in a higher $OC_{bio}$ contribution of 88%-91%, consisting of 43%-55% for $OC_{ts}$ and 36%-46% for
$OC_{lps}$. The small difference in source contribution mainly results from the fact that old $OC_{bio}$ from loess-paleosol
sequences was not considered in Yu et al. (2019a), and from the different isotopic signatures chosen for the POC
endmembers. However, both estimates ignore the possible presence of rock-derived OC in soils. In any case, our
results suggest that the Huanghe transports more $OC_{bio}$-derived POC than previously thought, with more aged, soil-
derived OC.
It is worth noticing that these calculations suggest that the $OC_{lps}$ fraction in the Huanghe was significantly higher in
2013 than in 2016. As most Huanghe sediments are derived from the CLP, higher physical erosion in the CLP should
enhance supply of aged, refractory $OC_{bio}$ to the river system. Consequently, the decrease in sediment supply from the
CLP initiated a few decades ago (Wang and Fu et al., 2016), which is likely to continue in the future, will probably
lead to the reduction of the contribution of $OC_{lps}$ to total POC export from the Huanghe. This might have an impact
on the burial efficiency of riverine POC on the continental margins, as $OC_{ts}$ is more labile than $OC_{lps}$, and thus more
prone to the remineralization process before burial. Moreover, decreasing erosion rate in the Huanghe basin will lead
to decreasing sediment accumulation rate in the estuary, which potentially favors the oxidation of all POC components
before burial (Blair and Aller, 2012). The time scale over which such effect could take place is yet unknown, as
anthropogenic intervention is the primary reason for the sediment yield reduction, through afforestation and soil and
water conservation measures in the CLP and reservoir operation in the middle reaches of the Huanghe. However, it is
plausible that in response to decreased terrestrial physical erosion on the Loess Plateau over at least decadal timescales,
an increased proportion of Huanghe POC will be oxidized before burial in the ocean, thereby leading to a weakened
preservation efficiency for the terrestrial eroded POC.

## 5.3 POC export by the Huanghe

In the Huanghe, POC content varies both vertically and laterally throughout the cross-section (Figure 2). This spatial
variability of both physical and chemical SPM characteristics must be considered when estimating integrated
instantaneous POC concentration and flux (Section 3.4).
We calculate that (Text S1) at the sampling time (July 2016), the Huanghe at Luokou transported 1,075 kg/s of SPM
for a water discharge of 731 $m^3$/s, such that the spatially-integrated SPM concentration over the cross section ($SPM_{int}$)
was 1,472 mg/L, a value relatively close to the straightforward average concentration of our 10 samples (1,286 mg/L).
The Luokou gauging station records a monthly SPM load of 1,826 kg/s in July 2016, and the daily average SPM load
of 1,096 kg/s for a daily average water discharge of 643 $m^3$/s on the 16[th] and 17[th] of July 2016 (method: three water
samples collected at 0.5 m below the channel surface across the transect profile, data available at
http://www.yrcc.gov.cn). Even though the latter estimate neglects the vertical heterogeneity within this relatively
shallow river (< 5.0 m), estimates give similar results.
We further obtain an instantaneous POC flux of 3.69 kg/s, corresponding to a cross-section integrated average POC
content ($POC_{int}$%) of 0.34% when dividing this instantaneous POC flux by the instantaneous SPM load. Given the
relatively homogenous distribution of $OC_{petro}$, the instantaneous flux of $OC_{petro}$ was calculated by multiplying the
average $OC_{petro}$ content by the instantaneous cross-section integrated SPM flux, yielding 0.44 ± 0.18 kg/s. The
instantaneous $OC_{bio}$ flux was then calculated by subtracting the instantaneous flux of $OC_{petro}$ from the instantaneous
POC flux, yielding 3.25 ± 0.20 kg/s. Assuming that our SPM samples are representative in terms of POC content
exported in July 2016, and taking the SPM flux of the gauging station for July, then the estimated fluxes of POC,
$OC_{bio}$, and $OC_{petro}$ for the flood period of July 2016 are 6.1, 5.4, and 0.7 kg/s, respectively. Taking $POC_{int}$ content for
estimating the annual POC flux yields a value of 1.1 kg/s consisting of 1.0 and 0.1 kg/s for $OC_{bio}$ and $OC_{petro}$ fluxes,
respectively. Note that these numbers are lower-bound estimates because POC content in Huanghe SPM collected
during flood periods is generally the lowest (Ran et al., 2013). Taking the highest POC content 0.75% reported in the
lower Huanghe in 2016 (Yu et al., 2019a), the estimated annual POC flux is 2.4 kg/s.
The above numbers present a sharp decrease compared to the estimated POC and $OC_{bio}$ fluxes transported by the
Huanghe over the period 2008 to 2013. Galy et al. (2015) estimated an $OC_{petro}$ flux of 1.9 kg/s and an $OC_{bio}$ flux of
11.4 kg/s from 2008 to 2012 (SPM flux: 3,655 kg/s, YRCC 2016), while Tao et al. (2018) reported an $OC_{petro}$ flux of
5.8 kg/s and a similar $OC_{bio}$ flux of 12.6 kg/s from June 2012 to May 2013 (SPM flux: 5,723 kg/s, YRCC 2016).
We first note that previous estimates of POC flux in the Huanghe might be biased as these estimates neglect the
variability over the cross-section (*e.g.,* Hu et al., 2015; Ran et al., 2013; Tao et al., 2015), SPM samples analyzed so
far for the Huanghe were generally collected within the first 0.5 m below the surface, meaning that previous POC
estimates did not consider the observed vertical and lateral POC heterogeneities and have thus misestimated POC
sources and fluxes. Those estimates were calculated by multiplying an individual surface POC content by the
corresponding monthly or weekly suspended sediment load, as provided by hydrological stations. Such estimates can
be problematic because POC content in SPM generally decreases from top to bottom (Figure 2), resulting in biased
surface-based estimates of fluxes (Bouchez et al., 2014). Using our cross-section data, we can estimate the bias in
POC flux estimates when a single sample is used for such flux estimates, by multiplying depth-integrated sediment
flux by the POC content of each sample. Such calculation shows different POC fluxes ranging from −15% to +30%
compared to the depth-integrated estimate, which is mostly influenced by the variable POC content. Considering SPM
collected at the channel surface, POC flux estimates using samples JB 1-1 and JB 2-1 are 28% and 15% higher,
respectively, and are 6% and 5% lower using samples JB 3-1 and JB 5-1, respectively, than the depth-integrated
estimate. This simple sensitivity analysis shows that channel surface sampling of SPM alone does not necessarily
result in an overestimation of POC flux because of lateral heterogeneity, even though the POC content of SPM is
generally higher at the surface than at the bottom (Figure 2). Consequently, and although accurate estimation of POC
fluxes requires grain-size variations to be accounted for, the corresponding bias cannot explain the large difference
between our estimates of Huanghe POC export for the year 2016 and previous estimates for preceding years. The SPM
flux is 336 kg/s in 2016 and 762 kg/s over the period 2014 to 2016, which is one order of magnitude lower than values
reported from 2008 to 2013 (YRCC 2016). The dramatic decrease in sediment load of the Huanghe (Wang and Fu et
al., 2016) has most likely exerted a first-order control on the reduction in POC export from the Huanghe river system,
and will probably continue to do so in the near future.
In the lower Huanghe, the POC content is very low and has small variance among different size fractions (Get et al.,
2020), such that the POC flux is controlled by the SPM flux. In particular, it is worth noting that the Huanghe displays
strong density stratification effects compared to other rivers (Moodie et al., 2022), with near-bed flow dominating the
transport of SPM. In order to appraise how spatial and temporal variability in SPM flux could influence POC export,
"local" POC loads can be calculated throughout the cross-section using the local water velocity, SPM concentration,

and POC content (Figure 8). In general, in the lower Huanghe more POC is transported near the riverbed and above shallower bathymetry on the left side of the channel (except for profile JB 5). For instance, there is a nearly two-fold increase in POC export from the surface to the bottom for the JB 2 and JB 3 profiles. The maximum local bulk POC (14.0 gC $m^{-2}$ $s^{-1}$), $OC_{bio}$ (12.2 gC $m^{-2}$ $s^{-1}$) , and $OC_{petro}$ export (1.8 gC $m^{-2}$ $s^{-1}$) are of sample JB 3-2, representing over 6 times the size of the corresponding minimum value (sample JB 5-1). This spatial pattern of POC load is almost the reverse of the POC% variation over the cross-section, again stressing the importance of sediment river dynamics in POC delivery. From these considerations, it could be anticipated that during the low-water season, when water velocity is slower, near-bottom Huanghe SPM is deposited on the channel bed, withdrawing a significant fraction of the POC export, as shown in other large rivers (Ke et al., 2022). This topic should be further examined in future research, in order to systematically investigate the stratification of sediment and associated OC transport dynamics in lowland and high-turbidity fluvial systems.

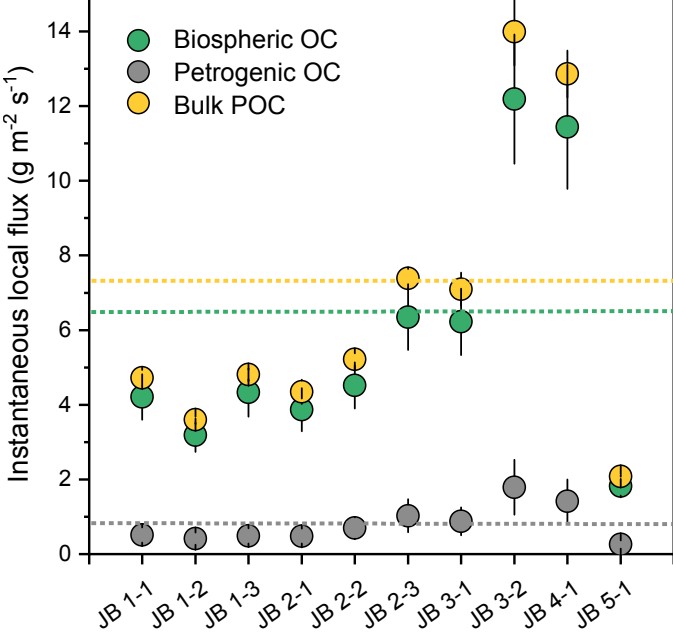

**Figure 8: Estimates of instantaneous "local" fluxes of Huanghe bulk POC, $OC_{bio}$ and $OC_{petro}$, calculated for each sample of the Luokou cross-section. The three dotted lines marked in orange, green, and grey represent the corresponding instantaneous, cross-section integrated fluxes. The error bar represents 1 standard deviation.**

Interestingly, anthropogenic activities may have antagonistic effects on POC export. Deforestation, agriculture, and mining have considerably enhanced the sediment yield from the CLP since the mid-Holocene (He et al., 2006) while the construction of large dams, soil and water conservation measures, and afforestation has considerably reduced the sediment yield since the 1950s (Wang and Fu et al., 2016; Wang et al., 2007; Syvitski et al., 2005). Yet the Huanghe exports substantial $OC_{bio}$ and $OC_{petro}$ with a significantly higher burial efficiency (avg. ca. 42%; Sun et al., 2018) than other large fluvial systems entering passive continental margins, such as the Changjiang, Amazon, and Mississippi (Blair and Aller, 2012). It is reported that aged soil OC is nearly fully preserved in continental margins and that $OC_{petro}$ has a *ca.* 70% burial efficiency (Tao et al., 2016). However, the contribution of the Huanghe OC burial to the global C sink is likely to be lower in the future as the consequence of 1) sharp decrease in SPM and POC export due to

weakened physical erosion in the CLP; 2) reduced sediment accumulation rate favoring OC remineralization in
estuaries (Blair and Aller, 2012; Walling and Fan, 2003; Milliman and Farnsworth, 2011; Galy et al., 2015).
**6 Conclusions**
In this contribution, we present the first detailed study of particulate organic carbon (POC) over a complete river cross-
section of the Huanghe, providing new perspectives on the transport mode, source, and instantaneous fluxes of POC
in this highly turbid large river.
At the scale of a cross-section, physical and chemical properties of SPM are heterogeneous both vertically and
laterally, a feature that is mainly controlled by bathymetry and hydrodynamic sorting. Resuspension of bed sediment
and local erosion of the right bank together impact the suspended POC composition at the sampled location. This
spatial heterogeneity shows that near-bottom SPM plays a dominant role in the delivery of $OC_{bio}$ (topsoil and deep
soil OC combined) and $OC_{petro}$. Despite a relatively shallow river channel ($< 5.0$ m) and narrow width ($< 200$ m), we
show how the heterogeneity of POC transport over a cross-section needs to be considered in constraining POC
transport mode and estimating POC fluxes.
Despite its millennial age, POC in the Huanghe is dominated by $OC_{bio}$ with a contribution of $88.0 \pm 1.3$ %. $OC_{petro}$
content in SPM is relatively homogeneous (0.04% - 0.05%) over the cross-section, indicating that the variability in
bulk POC age is mainly controlled by the variability in $OC_{bio}$ content, especially in the finest SPM fraction. $OC_{bio}$ ages
deduced from the application of a mixing model to previously published data (record period 2011-2016) are highly
variable, ranging from 1,779 to 8,325 [14]C yrs. We interpret this feature as resulting from the erosion of deep horizons
by gully systems in the loess-paleosol sequences containing [14]C-dead $OC_{bio}$. Enhanced erosion of deep loess-paleosol
horizons mobilizes aged and refractory OC to the ocean, with high burial efficiency on the passive margin. The erosion
of loess-paleosol horizons is thus an efficient process of $CO_2$ burial. However, the construction of large dams has
drastically affected the sediment load of the Huanghe system and retains substantial quantities of sediments that were
previously exported to the ocean. Future work is needed to further quantify how these anthropogenic modifications
alter POC composition and transport, by conducting comprehensive cross-section sampling campaigns over extended
time series upstream and downstream from dams.
**Appendix A**
Fluvial POC delivered in the Huanghe POC could originate from three terrestrial sources (Table. 2). As topsoil
typically contains recently photosynthesized $OC_{bio}$, we used a $\delta^{13}C$ value of $-24.8 \pm 1.9$‰ (n=166) according to the
subsurface soil OC values measured across the Huanghe basin (Rao et al., 2017). Over the sampled cross-section, the
depleted [13]C values indicate the dominant and almost exclusive input of C3 plant-derived material to the Huanghe
POC in the lower reaches. Based on [14]C (Liu et al., 2012) and [10]Be (Zhou et al., 2010) dating of $< 10$ cm-deep soil
horizons in the Huanghe Basin, the average age of topsoil was chosen as being younger than 2,000 yrs (*i.e.*, $\Delta^{14}C >$
$-220$‰). As for the topsoil end member includes modern biospheric material ($\Delta^{14}C$ around 40‰, Hua et al., 2013),
we assigned a $\Delta^{14}C$ value of $-90 \pm 130$‰ (Fm = $0.91 \pm 0.13$). This range also includes the range of $\Delta^{14}C$ values of
pre-aged soil OC indicated by the long-chain n-$C_{24+26+28}$ alkanols of the Huanghe POC reported by Tao et al. (2015)
and Yu et al. (2019a). Their results show consistent POC $\Delta^{14}C$ values in the lower reaches of $-204 \pm 20$‰ (Fm = 0.80
$\pm 0.03$, n=7) from June 2015 to May 2016 and $-219 \pm 33$‰ (Fm = 0.79 ± 0.04, n=4) at Kenli and of $-198\pm15$‰ (Fm
= 0.81 ± 0.02, n=6) from June 2015 to April 2016 at Huayuankou (Tao et al., 2015; Yu et al., 2019a).
The second end member should be characterized by aged and refractory OC from the loess-paleosol sequence
excluding topsoil (upper 10 m) of the CLP. Radiocarbon dating has an upper age limit of around 50,000 yrs, age above
which $F_m$ is equal to 0. However, radiocarbon-free OC spanning from 50,000 to 100,000 yrs must still be considered
as $OC_{bio}$ in the long-term carbon cycle. Here, we name this ignored OC as the "dormant" OC, without which the $OC_{bio}$
(*i.e.*, less than 100,000 yrs old) would be underestimated to some extent because the radiocarbon-free OC would be
misinterpreted as having a petrogenic origin. To consider this "dormant" OC, a $\delta^{13}C$ values of $-22.7 \pm 1.0$‰ (n=34,
Ning et al., 2006) and a $\Delta^{14}C$ values of $-610 \pm 390$‰ (Fm, 0.39 ± 0.39) were adopted based on an average of values
over the whole loess-paleosol sequence. Although radiocarbon-free (*i.e.*, older than 100,000 yrs) OC overlaps with
this end member, such old soil organic carbon is probably not mobilized as modern gully erosion mainly concerns the
upper 10 m of the loess-paleosol sequences, where soil $OC_{bio}$ is assumed to be significantly younger comparatively
(Figure 5).
Rock-derived OC from the QTP and the CLP, as well as kerogen from oil-gas fields from the Ordos Basin,
were all considered to be possible contributors to the $OC_{petro}$ endmember. The $\delta^{13}C$ of $OC_{petro}$ greatly varies between
the QTP ($-21.2 \pm 1.2$‰, n=11, Liu et al., 2007) and the CLP ($-26.8 \pm 0.5$‰, n=8, Qu et al., 2020). However, most of
the sediments eroded from the QTP are not transferred to the lower reaches as they remain trapped in the CLP and the
western Mu Us desert (Nie et al., 2015; Licht et al., 2016; Pan et al., 2016). In addition, the construction of large dams
in the upper reaches has considerably reduced the transfer of solid materials downstream (Wang et al., 2007).
Therefore, rock-derived OC inherited from the denudation of the QTP region is not further considered. Kerogen from
the oil-gas fields of the Ordos Basin in the CLP region (Figure 1) has $\delta^{13}C$ values of $-29.2 \pm 0.9$‰ (n=10, Guo et al.,
2014). Taking these constraints together, we consider a $\delta^{13}C$ value of $-28.1 \pm 1.5$‰ (n=18) for the $OC_{petro}$ end member,
and a $\Delta^{14}C$ value of $-1000$‰ (Fm = 0) by definition.
**Key Points**
• Bank erosion in lower Huanghe provides recent organic carbon to fluvial transport, altering the particulate

638        organic carbon transport over a river channel cross-section;

• Erosion of deep soil horizons of the loess-paleosol sequence contributes radiocarbon-dead organic carbon

640        from the biosphere to the Huanghe;

• Channel-bottom transport in the Huanghe is the primary process of exporting fluvial particulate organic

642        carbon to the estuary.

**Data availability**
All datasets are included in the paper and the supplementary materials.

**Author contributions**

DC, JB, CQ, and YK conceptualized the study. DC, JB, YK, MM, and BC determined the methodology. HC and JC collected the sediment samples. YK, MM, and AN assisted with elemental and isotopic carbon analysis. DC and CQ supervised the work. KY performed data analysis and wrote the original draft, and all authors contributed to the review and editing of the paper.

**Competing interests**

We declare there is no competing interest.

**Acknowledgments**

We thank Yulong Liu and Shengliu Yuan for their help during sampling and filtering. We also thank François Thil and Nadine Tissenerat for invaluable help when running the ECHoMICADAS, and Pierre Barré for use of the Beckman Coulter's LS 13 320 for paticle size analysis at École normale supérieure.

**Financial support**

This study was financially supported by the Agence Nationale de la Recherche (ANR) SEDIMAN (Grant ANR-15-CE01-0012), the National Natural Science Foundation of China (NSFC), grants 41561134017, 41625012, and the China Scholarship Council (CSC) to Yutian Ke (No.201706180008).

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
