# Peer review of "Channel cross-section heterogeneity of particulate organic carbon"

_EGUsphere, 2023_

## Author Comment (AC1)

Reviewer 1

This study investigating the source, transport, and fate of particulate organic carbon (POC) in the Huanghe, contributes to the comprehensive understanding of the global carbon cycle. The cross-channel sampling scheme used to investigate POC dynamics is novel and the identification of vertical heterogeneity in organic carbon transport, controlled by bathymetry and hydrodynamic sorting, is noteworthy. Indeed, it underlines the need for considering the heterogeneity of POC transport across channel sections while estimating POC fluxes and determining transport modes.

The authors provide an intriguing perspective on POC sources by using carbon isotopes (13C and 14C), suggesting the mobilization of aged and refractory organic carbon from the deeper soil layers of the loess-paleosol sequence in the Chinese Loess Plateau is a significant contributor to fluvial POC. The paper further presents a comparison of the calculated POC fluxes with existing literature, noting a significant reduction in POC flux in 2016 compared to that in the period of 2008-2013. This reduction, including OCbio and OCpetro, has been attributed to anthropogenic activities, mainly dam construction. However, there are some points I wish to raise for the authors' consideration.

Dear Reviewer, thank you for your positive comments. We have addressed all your concerns below and made corresponding changes in the manuscript.

First, the isotopic endmember values used for source deciphering are somewhat unclear, especially the topsoil's 13C endmember value of -24.8 ± 1.9‰, as the observed POC 13C range is -25 to -27‰ during the study period, which falls within the uncertainty of the endmember.

Rao et al., 2017 reported a large database of soil OC, we extracted the values for samples collected within the Huanghe basin, and a total of 166 samples were used to calculate an average value for the carbon isotopic composition of the topsoil organic carbon. The way we calculated the isotopic composition of endmembers is fully described in Appendix A. The $\delta^{13}$C of the topsoil OC endmember and POC samples clearly overlap but they have contrasting Fm values. This contrast helps separating topsoil OC contribution from total OC as both isotopic systems are used in the mixing model.

Secondly, the methods used to calculate POC fluxes, including those of OCbio and OCpetro, appear to be somewhat vaguely described (lines 470-490). I guess the authors used the observed instantaneous fluxes combined with the total suspended solids (TSS) from the gauging station to calculate monthly fluxes, which were then extrapolated to annual fluxes. And then the annual mean flux in per second unit was obtained for comparison. However, the error introduced by converting one snapshot to the annual averaged flux needs to be explicitly addressed to support their statement regarding the POC flux reduction. To my knowledge, there hasn't been a significant reduction in sediment discharge since 2008.

Thanks for the suggestion. We also realized estimating the annual flux by constraining the lower band is not sufficient. We further estimated the upper band of the annual flux by the highest POC content reported in the lower Huanghe (Yu et al., 2019a), which yields a POC flux of 2.4 kg/s. This value is still one order of magnitude lower than the values reported in Galy et al., 2015 and Tao et al., 2018.

Besides, a significant reduction in sediment load has been observed since 2008 even though the sediment load was overall already low since dam operation. Indeed, the sediment load in 2008 has been estimated to 2,442 kg/s while it was around 336 kg/s in 2016, so a decrease of about one order of magnitude. We now provide in the main text additional numerical information on the sediment flux in the lower Huanghe to strengthen our statement.

"The SPM flux is 336 kg/s in 2016 and 762 kg/s over the period 2014 to 2016, which is one order of magnitude lower than values reported from 2008 to 2013 (YRCC 2016)."

Overall, this manuscript provides valuable insights into POC transport, sources, and instantaneous fluxes, potentially advancing the current understanding in the field. With further elucidation of the points raised, it could be an important contribution to the literature.

Reviewer 2
This study presents a detailed examination of particulate organic carbon dynamics in the lower Huanghe River, a highly managed river system. Through a comprehensive analysis of sediment and particulate organic carbon concentrations and compositions across a channel transect, distinct patterns are observed both laterally and vertically, which can be attributed to the influence of riverine hydrodynamics. To quantify the contributions from different sources, a dual carbon isotope mixing model is applied, considering inputs from topsoils, the Chinese Loess Plateau, and petrogenic carbon. Additionally, the study utilizes a Rouse model to simulate instantaneous fluxes. These findings contribute to our understanding of POC dynamics in the lower Huanghe River and its complex interactions with hydrodynamics, providing valuable insights for the management and conservation of riverine ecosystems.

The study emphasizes the importance of conducting depth sampling across a river transect to precisely assess sediment and particulate organic carbon concentrations. It acknowledges the significant impact of hydrodynamics on the distribution and composition of both organic and inorganic components within the water column. Moreover, accurate modeling of export fluxes relies on a comprehensive understanding of these factors. The manuscript is written in suitable language and aligns well with the scope of Earth Surface dynamics.

However, the manuscript contains minor disconnections between the presented data and the corresponding interpretations. To enhance the discussion section, it would be valuable to provide a more detailed and thorough analysis of the hydrodynamic mechanisms, particularly in relation to the Rouse model. This detailed analysis would provide valuable insights into the observed heterogeneity in biogeochemical and sedimentary characteristics within the depth samples. Additionally, expanding the literature review to encompass relevant studies and recent publications that utilize river depth sampling would further enhance the manuscript's robustness and overall quality, by placing the findings within a broader scientific context.

The manuscript lacks sufficient provision and explanation of the statistical metrics used in the analysis, which undermines transparency and reproducibility. To address this concern, it is crucial to offer a more thorough explanation of these metrics, ensuring that readers can understand and replicate the analysis with clarity.

A major concern arises from the incomplete reporting of the Bayesian mixing model, which limits reproducibility. The model description lacks essential details, such as parameterization, prior and posterior distributions, and convergence diagnostics. Detailed guidelines are provided below to address these concerns and improve the reporting of the model.

Before recommending the publication of this study, it is crucial to address these shortcomings and fully resolve the concerns that have been raised.

Dear Dr. Schwab,

Thank you for the comprehensive and valuable comments, we improved the manuscript by addressing all your concerns, in particular, we strengthened the mixing model, statistical validation and discussion section.

Lines 18-20: This statement does not accurately apply to JB-1-3, as the highest radiocarbon values are observed at maximum depth in this particular case. Rephrase this sentence more carefully.

We have rephrased the sentence.

Lines 142-143: Please provide the coordinates for the sample location.

The coordinates are now provided.

Lines 143-144: Please provide references to any relevant previous studies to support your statement.

For sampling site, we obtained the number from our own survey and the number is validated by data records available at YRCC (Yellow River Conservation Committee): Annual Sediment Report for the Yellow River, 2016. The reference has been added to the text

Lines 168-172: Kindly provide the established standards for both stable and radiocarbon measurements. Moreover, was the amount of extraneous carbon taken into consideration during the radiocarbon measurements?

The following information have been added to the "sampling and analytical methods" section:

For stable carbon isotope: Subjected to the blank subtraction by linearity test, two international standards including USGS-40 and IAEA-600 as well as an internal standard (GG-IPG) were used to build linear regression equations to calibrate the elemental and isotopic values for both carbon and nitrogen.

For radiocarbon: we followed the protocol reported in Hatté et al., 2023 at LSCE, extraneous carbon was considered. Aside from the gas bottles of prepared blank PhA and standard NIST OX II which are permanently connected to the GIS and, used for normalization and corrections for fractionation and background, international standards including IAEA-C5, IAEA-C7, IAEA-C8, and blank PhA were prepared in different sizes (10 to 100's μg C) to match the amount of OC found in the sediment samples.

Hatté, C., Arnold, M., Dapoigny, A., Daux, V., Delibrias, G., Boisgueheneuc, D. D., Fontugne, M., Gauthier, C., Guillier, M.-T., Jacob, J., Jaudon, M., Kaltnecker, É., Labeyrie, J., Noury, C., Paterne, M., Pierre, M., Phouybanhdyt, B., Poupeau, J.-J., Tannau, J.-F., Thil, F., Tisnérat-Laborde, N., and Valladas, H.: Radiocarbon dating on ECHOMICADAS, LSCE, Gif-Sur-Yvette, France: new and updated chemical procedures, Radiocarbon, 1–16, https://doi.org/10.1017/RDC.2023.46, 2023.

Lines 174- 183: Since you have implemented a custom Bayesian approach instead of utilizing a reported R package, it is essential to provide additional information in either Section 3.3 or Appendix A, expounding on the Bayesian modeling methodology. I recommend to include the following specific details:

1. Data Variables: Clearly specify the variables used in the analysis and their corresponding data sources.

2. Likelihood Function and Parameterization: Describe the likelihood function employed in the Bayesian model and provide details on how the model parameters are parameterized.

3. Prior Distribution: Clearly state whether the prior distribution used for the model is informed or non-informed. Provide a formal specification of the prior distribution.

4. Prior and Posterior Predictive Checks: Explain the procedures followed for conducting prior and posterior predictive checks, which involve comparing model predictions with both observed and simulated data.

5. Model Comparison: Describe the approach used for comparing different models and evaluating their relative performance in terms of predictive accuracy.

6. Model Bias (Geometric Surface Area): Elaborate on how the concept of model bias, specifically related to geometric surface area, was incorporated and assessed in the analysis.

7. Software Used: Specify the software or programming environment employed for implementing the Bayesian modeling approach.

In addition, it is important to assess model convergence using diagnostic tests such as Geweke, Gelman-Rubin, and Heidelberg-Welch. These tests provide valuable insights into the behavior of Markov Chain Monte Carlo (MCMC) chains and ensure reliable results. To gain further guidance on best practices for mixing models, consider referring to Phillips et al. (2014, Canadian Journal of Zoology) and Kruschke (2021, Nature Human Behavior). Furthermore, it is recommended to carefully evaluate the burn-in period in relation to the length of the MCMC chain to address concerns about convergence adequacy. A short burn-in period may raise questions about convergence. Lastly, it is

crucial to provide accurate and comprehensive reporting of the posterior analysis in either the Results or Discussion section, enabling readers to reproduce the analysis effectively.

Thanks for the comprehensive suggestion, we also noticed that the citation for using this method is not complete. We do utilize a reported R package, which is *rjags*, a program for analysis of Bayesian hierarchical models using Markov Chain Monte Carlo (MCMC) simulation, available at https://cran.r-project.org/web/packages/rjags/index.html.

The text in the method section 3.3 (POC source apportionment) has been profoundly rewritten as follows:

 "To quantify the contribution and associated uncertainties of various sources to POC transported in the Huanghe, a Bayesian Markov Chain Monte Carlo (MCMC) based on a three-end member (Appendix A) mixing scheme was adopted. This approach considers the variability on each end member contribution, assuming this variability can be represented by a normal distribution. Prior information is assumed to be unknown. We computed the posterior distribution of the Bayesian formulation using the MCMC method, facilitated by the *rjags* package (https://cran.r-project.org/web/packages/rjags/index.html, Andersson et al., 2015), all computations were performed in the R environment (http://www.r-project.org/). To ensure reliable simulation, the model was run with 5,000,000 iterations, using a burn-in of 1000 steps, and a data thinning of 100 for each sample."

Line 201: When reporting the average, the corresponding standard deviation or standard error should be included, along with the number of samples.

Thanks for having spotted this missing information. We now report the average value with 1 standard deviation (SD) in the manuscript.

Line 208: Figure S4 illustrates the values of D10 and D90. However, the actual data corresponding to these values has not been provided in the report.

Actually, the D10 and D90 values are reported in the figure S4, not in a table as for the D50.

We added "In each depth profile, SPM is consistently coarsening with depth as revealed by the evolution of grain size parameters such as D10, D50, and D90 (Table 1; Figure 2 and Figure S4)." in the main text.

Line 208-2010: Upon visual inspection, the suggested bimodal distribution of these sediment curves is hardly discernible.

Not all samples show bimodal distribution for grain size, we emphasize that some of the samples show this pattern, such as JB 4-1, 1-1, 1-2, 1-3.

Lines 2018-2019: To avoid confusion, it is necessary to clarify whether the reported value represents the standard deviation or the standard error. Prior to utilizing the value, please provide a clear definition of the specific measure being used.

Given that the extreme Al/Si values are listed in the main text, we removed the average value from the main text as it did not provide more information.

Lines 295-297 and 317-324: To gain insights into the observed variation in the Huanghe River (JB-1-3), it would be beneficial to reference recent studies that have provided evidence for the systematic transfer and export of discrete plant-derived debris above the riverbed in major river systems. Consider examining research papers such as Feng et al. (2016, JGR: Biogeosciences), Lee et al. (2019, PNAS), and Schwab et al. (2023, JGR: Biogeosciences) for a better understanding of this phenomenon.

Lines 320-323: Given that your sampling focuses on high stage conditions, it is important to acknowledge that the inundation of adjacent riparian zones may contribute to the mobilization and entrainment of discrete plant-derived debris. It is advisable to also take into account factors such as surface runoff triggered by storm events and direct litterfall. These additional considerations can provide insights into the dynamics of plant-derived debris mobilization in the study area.

Here we address all comments posed for Lines 295 -324 to integrate the mechanism of how plant debris get exported to the channel and its transport mode in large rivers. We extended the discussion as follows.

"Bank erosion can be a significant mechanism for the delivery of sediments to river systems (Guo et al, 2007). Bank erosion at Luokou would make OC from the lower Huanghe alluvial plain a potential source of POC in the lower reaches of the Huanghe. Frequent inundation to the adjacent riparian zones in flooding seasons, surface runoff driven by storm events, and agriculture irrigation etc., can mobilize young soil OC and discrete organic matter debris (e.g., plant-derived debris) to riverine POC (Hilton et al., 2011; Turowski et al., 2016). This mechanism provides a possible explanation for the opposite trends displayed by samples from the JB 1 and JB 2 profiles in the Fm vs. δ13C space (Figure 5). The youngest POC was found at the bottom of the JB 1 profile (JB 1-3). Meanwhile, the JB 1 samples have comparatively higher N/$C_{org}$ ratios and N%, consistent with the input of discrete plant-derived debris from the bank in addition to rock-derived detrital clastic material in the coarse fractions (> 32 μm, Yu et al., 2019b). The transport and entrainment of plant debris deep in the water column has been evidenced in many large river systems, such as the Amazon (Feng et al., 2016), the Mackenzie (Schwab et al., 2022), the Rio Bermejo River (Repasch et al., 2021). Such input would also provide an explanation for the higher POC loading of the JB 1 profile (Section 5.1.1)."

Lines 309-312 and 325-332: To enrich the interpretation and foster a more comprehensive understanding of the chemical composition variations within the transect, it is valuable to reference your Rouse model. This model, characterized by its ability to offer a more continuous representation, can effectively shed light on the impact of hydrologic dynamics, such as gravitational settling and resuspension. By incorporating a discussion based on the Rouse model, a deeper understanding of how these hydrologic processes influence the observed variations in chemical composition can be achieved within the transect.

This is a very good suggestion; we now discussed the Rouse number and the Rouse model. Please check the updated discussion below:
"At the Luokou sampling site, lateral variability at the channel surface shows that POC-rich fine particles are preferentially transported near the right bank (Figure 2 and Figure S3). This pattern is validated by the Rouse model provided in Text S1, the Rouse number ($Z_R$) is 0.137, 0.236, and 0.284 for JB-1, JB-2, and JB-3, respectively. In essence, the Rouse number ($Z_R$) can reflect the balance between gravitation settling and upward turbulent diffusion. $Z_R$ is smaller near the right bank while larger near the left bank, showing heterogeneity across the transect. Larger particles exhibit a faster settling velocity due to their increased weight, leading to a higher $Z_R$. On the other hand, the lighter ones settle more slowly, resulting in $Z_R$ approaching 0. This means that their concentration remains relatively constant along a given depth profile. However, as depth increases and the concentration of larger particles grows, the proportion of these finer particles in the overall sediment decreases (Bouchez et al., 2011a). The channel geometry thus needs to be examined as a potential factor to produce such lateral heterogeneity, in particular the mechanisms of bed sediment resuspension and bank erosion.

Resuspension of bed sediments is also a possible mechanism that could explain the lateral heterogeneity in POC content in the study cross-section of the Huanghe. Indeed, scouring of channel bed sediment at high water flow may also shift POC to more negative radiocarbon and stable isotope signatures. Our sample set collected in July 2016 during a flooding period (water flow velocity up to 2.1 m/s, Figure 1) supports this scenario. Indeed, the increase in D50 of surface SPM samples from right to the left bank, that is with total channel depth decrease, is consistent with coarse sediment resuspension from the bed. This is also supported by the Rouse model, where higher $Z_r$ in the shallow water near the left bank indicates a greater likelihood of sediment settling to the bed, lower $Z_r$ suggests that there is enhanced SPM supply from the riverbed. Such a scenario is also supported by the three-fold increase in SPM flux observed from the upstream Huayuankou station to the downstream Lijin station in July 2016, despite a four-fold decrease in water discharge (Figure S2)."

Lines 349-352: This sentence is ambiguous. Could you please provide further elaboration on how a reduction in sediment load can impact the radiocarbon composition? Additionally, it would be helpful to discuss the primary sources of radiocarbon that were prevalent before 1950.

We agree with this comment, and we therefore extended the discussion with "The Huanghe has experienced a 90% decrease in annual sediment load since the 1950s (Wang et al., 2015), caused by weakened soil erosion to the Chinese Loess Plateau and sediment retention by dams (Wang et al., 2007; Ran et al., 2013; Wang and Fu et al., 2016). To determine the contributions of the various terrestrial OC components to Huanghe POC, we compiled published POC carbon isotope data for sediments collected in the lower reaches from 2011 to 2016, after the Xiaolangdi Reservoir operated (Figures 5 and 6). This dataset shows that the radiocarbon ages of Huanghe POC are considerably old ($5,100 \pm 1,700$ $^{14}$C yr, n=29), with a minor fraction of modern photosynthesized $OC_{bio}$ (Tao et al., 2015; Yu et al., 2019a, b). This relatively $^{14}$C-depleted POC suggests the significant contribution of OC originated from deep soil horizons within the catchment. Given that loess is easily erodible and that there is widespread gully erosion in the catchment, more intensive erosion of the Chinese Loess Plateau can mobilize more soils as well as older OC from deeper soil horizons to the fluvial transport. Therefore, higher sediment load in the river can be characterized by radiocarbon-depleted POC. This is evidenced by the negative trend between $^{13}$C and Fm of POC for sediment samples collected in the Huanghe over the 2011-2016 period (Figure 5a), suggesting that deep horizons of the loess-paleosol formations are a plausible source for the $^{14}$C-depleted end member."

We also provide more discussion on this topic in the last paragraph of the section and talked about the soil erosion mechanism in the Chinese Loess Plateau in the Introduction section. Bomb carbon could also contribute enriched radiocarbon signals to the OC in the topsoils, thus further affecting POC in the Huanghe.

Lines 360-364: To strengthen your argument, consider including specific numerical values for the erosion rate of the Chinese Loess Plateau. This addition will provide quantitative support to your statement.

Thanks for the comments, we added the actual erosion rates of the Chinese Loess Plateau to the last paragraph in this section, where we are talking about gully erosion. "From 1925 to 1981, the erosion rate of the CLP was 6,318 t km-2 yr-1, compared to 10,770 t km-2 yr-1 in the hilly and gully plateau (Li et al., 2022). While the CLP's erosion rate dropped to 3,476 t km-2 yr-1 between 1982 and 2016, the rate in the hilly and gully plateau remained significantly high at 6,146.5 t km-2 yr-1 (Li et al., 2022)."

Lines 527-537: It is advisable to consider the potential effects of climate change-induced environmental changes. This intensified monsoon activity is expected to have a positive influence on the erosivity of the Chinese Loess Plateau, consequently impacting the transport of sediment to the Huanghe River.

This is a very good point, however, in this paragraph, we emphasized how anthropogenic activities affect sediment erosion within the Huanghe basin. We added this point to section 5.2.1, "Notably, the strengthening of the East Asian Monsoon in coming decades (Li et al., 2022; Xue et al., 2023) could potentially enhance this process."

Appendix A: In order to evaluate the robustness of the endmember composition, the numerical values of the endmember composition should be supplemented with the corresponding sample numbers. Additionally, it may be worth considering weighted means and standard deviations of each source relative to their respective carbon content. To enhance readability, I recommend presenting the endmember compositions in the form of a table. Furthermore, why was vegetation not considered as an endmember composition? Given that the sampling campaign took place during a high river stage, it may be crucial to account for the entrainment of plant-derived debris from the proximal floodplain due to flooding, as well as the direct litterfall.

Thanks for the critical suggestion, it would be hard to do the weighted means and standard deviations for each source since most of them did not report the OC wt.%. Besides, for some endmembers, stable carbon isotopic values and radiocarbon values are from two set of data from different sources, such as for topsoil. For these reasons, we will still use the average values and standard deviations to represent each endmember composition. We did not include vegetation as an endmember because: 1) when constrain topsoil as an endmember, it will overlap to a great extent with the isotopic values of vegetation in the Chinese Loess Plateau and the catchment, we provide this

explanation in the first paragraph in Appendix A; 2) it was reported in Tao et al., 2015 and Yu et al., 2019a, b using compound specific carbon isotopes, contribution of modern photosynthesized $OC_{bio}$ to fluvial OC in the lower Huanghe is few; 3) we did not observe any visible discrete plant-derived debris during pretreatment.

The table of the endmember compositions was added (Table 2) in the main text, please check it.

**Table 2: Summary of $\delta^{13}C$ and $\Delta^{14}C$ of source end members for POC in the Huanghe.**

| End member | $\delta^{13}C$ | $\Delta^{14}C$ |
|---|---|---|
| $OC_{ts}$ | −24.8 ± 1.9‰ | −90 ± 130‰ |
| $OC_{lps}$ | −22.7 ± 1.0‰ | −610 ± 390‰ |
| $OC_{petro}$ | −29.2 ± 0.9‰ | -1000‰ |

Figure 1: As sediment retention in dams is a significant aspect of your analysis, incorporating the dam locations on the map will provide vital visual information and enhance the understanding of sediment dynamics within the studied system.

Figures 3, 4, and 5: To bolster the connection between chemical properties and hydrological dynamics, it would be advantageous to incorporate a size parameter in your plots that reflects sampling depth, flow velocity, or discharge. By adding a third dimension to your plots, such as varying marker sizes, you can visually represent the additional hydrological information.

Figure 6: To ensure a thorough analysis, it is important to report further regression information, such as the equation, p-value, number of samples, root mean square error (RMSE), and mean absolute error (MAE). These details will provide a comprehensive understanding of the statistical relationship between the variables. Additionally, considering the incorporation of the particulate organic carbon content as a size parameter in your plots will enhance the visualization and enable the exploration of its potential impact on the observed patterns.

Thanks for the suggestions, we updated these figures.

For Figure 1, we added an extra layer of large dams in the Yellow River.

We used sampling depth to adjust the scatters in Figures 3, and 5. For Figure 4, we did not apply the third dimensional information, because we want to focus on the "POC loading" indicated by the relation between Al/Si ratio and POC%, and the data from each river is a compilation of different depth profiles.

For Figure 3 and Figure 6, we added p-value and RMSE. For Figure 6, we further indicated the number of samples, and marker size indicates POC wt. %. The further interpretation of added information was added in the corresponding places in the main text.

---

## Referee Report (RR1)

Dear Dr. Hilton and authors,

I appreciate the authors' efforts in addressing my initial comments, but there are still some crucial recommendations for enhancing scientific transparency and reproducibility that were overlooked. The comments below pertain to the tracked changes manuscript.

Lines 204 to 217:

1. The geometric surface area (GSA) is not reported. I recommend using the calc_area algorithm from the MixSIAR R-package (Brett, 2014; Stock et al., 2018) to compute GSA and test if the end-member composition is distinct for all end-members. The GSA value should be included in the manuscript.
2. Please specify which convergence diagnostic tests were employed for estimating model convergence (Geweke, Gelman-Rubin, or Heidelberg-Welch).
3. Report the chain length and the number of chains used in the analysis.
4. The burn-in of 1000 appears to be relatively small for an unmixing model; typically, it would be in the range from 100,000 to 1,000,000.
5. In Table 2, the average values for the endmembers should be accompanied by the number of samples used to calculate the mean $\delta^{13}C$ and $\Delta^{14}C$ for each. Report the amount of samples used for calculating $\delta^{13}C$ and $\Delta^{14}C$.

To adhere to APA statistical guidelines for scientific work, it is essential to cite results properly. Each mean should be reported with either a standard deviation (SD) or standard error (SE) or a similar metric. Indicate with the first usage of the mean and error statistic whether it is (M ± SD) or (M ± SE). M, SD, and SE are accepted APA abbreviations. This essential information should not be excluded for the reader's clarity, and I highly recommend including the sample size (n) for each average.

- Line 241: Missing SD and n, and lack of proper description.
- Lines 259-260: Missing n.
- Line 264: Missing n.
- Line 266: Missing n.
- Line 270: Missing n, and missing SD and n for the $\Delta^{14}C$ value.
- Line 272: Missing SD and n.
- Lines 274-275: Missing SD.
- Line 307: Missing n.
- Lines 615-617 and following: Missing SDs and ns.

Line 247: All used data should be made accessible to the reader. Please add the D10 and D90 values to Table 1, especially as this data is not deposited into a database. Reporting values in a figure is insufficient.

Line 310 and following, including figures: Use two or three decimal places and report exact values for all *p*-values greater than .001. For *p*-values smaller than .001, report them as p < .001 (APA).

Figure 1: Spelling error - "Guaging station" instead of "gauging station" within the map.

Figure 3 and 5: Add a legend for sample depth. Does the small dot represent surface or deep water?

Thank you for your attention to these matters.

Best regards,

---

## Author Response (AR2)

Reply #2 to reviewer

Dear Dr. Schwab and Dr. Hilton,

Please find bellow a point-by-point reply to the comments of reviewer Dr. Schwab.

*"Dear Dr. Hilton and authors,*

*I appreciate the authors' efforts in addressing my initial comments, but there are still some crucial recommendations for enhancing scientific transparency and reproducibility that were overlooked. The comments below pertain to the tracked changes manuscript."*

Thank you for your valuable input on this paper. Regarding the major concern about employing the Monte Carlo Model to delineate the contributions of different end members, I have incorporated your suggestion to use the MixSIAR package in the R environment. After conducting multiple validation calculations, I observed that this model yields results differing from previous reports. However, this discrepancy does not detract from the paper's main argument. On the contrary, the higher proportion of $OC_{lps}$ (organic carbon in loess-paleosol sequences) reaffirms our hypothesis that intensified erosion of deep loess-paleosol sequences is a significant contributor to fluvial particulate organic carbon (POC) in the Huanghe River. The manuscript has been revised in the following key areas:
1. Model: Implementation of the MixSIAR model, accompanied by diagnostic runs.
2. Figures: Updates to Figures 1, 3, 6, 7, and 8. Enhancements to the captions of Figures 3 and 5, including illustrations to denote that scatter size corresponds to collection depth.
3. Supplementary Material: Inclusion of Table S1 detailing particle size parameters and addition of Figure S7, which illustrates the geometric surface area in the carbon isotope mixing space."

*"Lines 204 to 217:*
*1. The geometric surface area (GSA) is not reported. I recommend using the calc_area algorithm from the MixSIAR R-package (Brett, 2014; Stock et al., 2018) to compute GSA and test if the end-member composition is distinct for all end-members. The GSA value should be included in the manuscript.*
*2. Please specify which convergence diagnostic tests were employed for estimating model convergence (Geweke, Gelman-Rubin, or Heidelberg-Welch).*
*3. Report the chain length and the number of chains used in the analysis.*
*4. The burn-in of 1000 appears to be relatively small for an unmixing model; typically, it would be in the range from 100,000 to 1,000,000.*
*5. In Table 2, the average values for the endmembers should be accompanied by the number of samples used to calculate the mean d13C and D14C for each. Report the amount of samples used for calculating d13C and D14C."*

Thank you for your comments. We have now implemented the MixSIAR model, with detailed information provided in Section 3.3.
GSA: We used the calc_area algorithm to get the GSA which is 7.13 $SD^2$ (provide in the caption of Figure. S7), the end-members are distinct.Model: "Prior information is assumed to be

uninformative. We computed the posterior distribution of the Bayesian formulation using the MCMC method, facilitated by the MixSIAR package (Moore & Semmens, 2008; Stock & Semmens, 2016), all computations were performed in the R environment (http://www.r-project.org/). To ensure reliable simulation, the model was run with chain length of 300,000 by 3 chains, using a burn-in of 200,000 steps, and a data thinning of 100 for each sample." Diagnostics: "Further model diagnostics was performed using Gelman-Rubin and Geweke test, both diagnostics validated the robustness and convergency of the model."

*"To adhere to APA statistical guidelines for scientific work, it is essential to cite results properly. Each mean should be reported with either a standard deviation (SD) or standard error (SE) or a similar metric. Indicate with the first usage of the mean and error statistic whether it is (M ± SD) or (M ± SE). M, SD, and SE are accepted APA abbreviations. This essential information should not be excluded for the reader's clarity, and I highly recommend including the sample size (n) for each average.*
*• Line 241: Missing SD and n, and lack of proper description.*
*• Lines 259-260: Missing n.*
*• Line 264: Missing n.*
*• Line 266: Missing n.*
*• Line 270: Missing n, and missing SD and n for the D14C value.*
*• Line 272: Missing SD and n.*
*• Lines 274-275: Missing SD.*
*• Line 307: Missing n.*
*• Lines 615-617 and following: Missing SDs and ns."*

Thank you for your thorough review. We have incorporated the suggested information. To enhance clarity and avoid redundancy, we explicitly state at the outset that our results and analysis rely on 10 samples collected from the cross section. We indicate in the result section that we always report M±SD (when necessary) in the manuscript.

*"Line 247: All used data should be made accessible to the reader. Please add the D10 and D90 values to Table 1, especially as this data is not deposited into a database. Reporting values in a figure is insufficient."*

We added Table S1 in the supplementary material to report these values.

*"Line 310 and following, including figures: Use two or three decimal places and report exact values for all p-values greater than .001. For p-values smaller than .001, report them as p < .001 (APA)."*

We adopted your suggestion, using 0.001 as the threshold.

*"Figure 1: Spelling error - "Guaging station" instead of "gauging station" within the map. Figure 3 and 5: Add a legend for sample depth. Does the small dot represent surface or deep water?"*

The spelling error in Figure 1 is corrected. For Figure 3 and 5, we added "The size of each circle indicates the water depth at the corresponding SPM, with smaller circles representing shallower depths and larger circles indicating deeper waters" in the caption.

Thank you for your attention to these matters.
Best regards,

Dr. Yutian KE, on behalf of all co-authors